

# Mineral formation induced by cable bacteria performing long-distance electron transport in marine sediments

**Nicole M.J. Geerlings[1], Eva-Maria Zetsche[2,3], Silvia Hidalgo Martinez[4], Jack J. Middelburg[1], Filip J.R. Meysman[4,5]**

[1]Utrecht University, Department of Earth Sciences, Princetonplein 8a, 3584 CB Utrecht, The Netherlands

[2]University of Gothenburg, Department of Marine Sciences, Carl Skottsberg gata 22B, 41319 Gothenburg, Sweden

[3]Department of Estuarine and Delta Systems, Royal Netherlands Institute for Sea Research and Utrecht University, Korringaweg 7, 4401 NT Yerseke, The Netherlands

[4]Ecosystem Management Research Group, Department of Biology, Universiteit Antwerpen, Universiteitsplein 1, 2160

Antwerpen, Belgium

[5]Department of Biotechnology, Delft University of Technology, Van der Maasweg 9, 2629 HZ Delft, The Netherlands

*Correspondence to*: Nicole M.J. Geerlings (N.M.J.Geerlings@uu.nl)



**Abstract**

Cable bacteria are multicellular, filamentous microorganisms that are capable of transporting electrons over centimeter-scale distances. Although recently discovered, these bacteria appear to be widely present in the seafloor, and when active, they exert a strong imprint on the local geochemistry. In particular, their electrogenic metabolism induces unusually strong pH excursions in aquatic sediments, which induces considerable mineral dissolution, and subsequent mineral re-precipitation. However at present, it is unknown whether and how cable bacteria play an active or direct role in the mineral re-precipitation process. To

this end we present an explorative study of the formation of sedimentary minerals in and near filamentous cable bacteria using a combined approach of electron microscopic and spectroscopic techniques. Our observations reveal three different types of biomineral formation directly associated with cable bacteria: (1) the formation of intracellular polyphosphate granules, which are associated with a localized accumulation of calcium and magnesium, (2) the attachment and incorporation of clay particles in a sheath surrounding the surface of the cable bacterium filaments, and (3) the encrustation of cable bacteria filaments by

newly formed solid phases containing high amounts of iron. These findings suggest a complex interaction between cable bacteria and the surrounding sediment matrix, and a substantial imprint of the electrogenic metabolism on mineral diagenesis and sedimentary biogeochemical cycling. Particularly the encrustation process leaves many open questions for further research. For example, we hypothesize that the complete encrustation of filaments might create a diffusion barrier and negatively impact the metabolism of the cable bacteria.

**Keywords:** Cable bacteria, biominerals, poly-phosphate granules, encrustation, electron microscopy, X-ray spectroscopy

# 1 Introduction

## 1.1 Cable bacteria

In 2012, long, multicellular micro-organisms were reported from in marine sediments that are capable of generating and mediating electrical currents over centimeter-scale distances (Pfeffer et al. 2012). These so-called "cable bacteria" have

evolved a unique energy metabolism, in which electrons are passed on from cell to cell, thus establishing an electrical current from one end to the other of their centimeter-long filamentous bodies. This biological innovation equips cable bacteria with a competitive advantage for survival within the redox gradients that exist within the seafloor (Meysman, 2017). The electrical connection allows electron donors and electron acceptors to be harvested in widely separated locations (Nielsen et al., 2010), and in this way, the energy yield from natural redox gradients can be favorably optimized.

The currently known cable bacteria strains belong to two candidate genera within the *Desulfobulbacea* family of the delta-proteobacteria: the marine "*Candidatus* Electrothrix" and the freshwater "*Candidatus* Electronema" (Trojan et al., 2016). Cable bacteria perform electrogenic sulfur oxidation (e-SOx) via long-distance electron transport (LDET). Thereby, they engender an electrical coupling of the oxidation of sulfide within deeper sediments with the reduction of oxygen near the





sediment surface (Fig. 1). The anodic half reaction in the anoxic zone (half-reaction: $\frac{1}{2}H_2S + 2H_2O \rightarrow \frac{1}{2}SO_4^{2-} + 4e^- + 5H^+$)

generates electrons, which are then transported along the longitudinal axis of the filament to cells located in the oxic zone (Pfeffer et al., 2012). In this thin oxic zone, cathodic cells consume the electrons via oxygen reduction (half reaction: $O_2 + 4H^+ + 4e^- \rightarrow 2 H_2O$). Field investigations have shown that this separation of redox half-reactions can span a distance up to 7 cm (van de Velde et al., 2016), thus illustrating the wide spatial scale over which LDET can be active.

LDET has recently been documented to be active in a wide range of marine environments (Malkin et al., 2014; Burdorf et al., 2017) , including salt marshes, mangroves, seagrasses, and seasonally hypoxic basins, as well as in freshwater streambeds (Risgaard-Petersen et al., 2015) and possibly in aquifer sediments (Müller et al., 2016). LDET has furthermore proven to exert a dominant influence on biogeochemical transformation and fluxes in the seafloor. This impact can even extend beyond the sediment, and can control the overall cycling of iron, sulfur and phosphorus within the sediment and water column of coastal

systems (Seitaj et al., 2015; Sulu-Gambari et al., 2016). The widespread occurrence of cable bacteria and their strong local geochemical imprint suggests that they could be important in the cycling of carbon, sulfur, iron and other elements in various natural environments (Nielsen and Risgaard-Petersen, 2015).

### 1.2 Acidification and mineral cycling

A remarkable aspect of the metabolism of cable bacteria is that they induce unusually strong pH excursions in aquatic sediments (Nielsen et al, 2010; Meysman et al, 2015). These large pH excursions are a direct consequence of the spatial segregation of redox half-reactions made possible by LDET (Fig. 1). Considerable amounts of protons are released by anodic sulfur oxidation in deeper sediment horizons, thus strongly acidifying the pore water (down to pH 6 and below). Oppositely, large quantities of protons are consumed by cathodic oxygen consumption, leading to a characteristic pH maximum in the oxic

zone (with pH values up to 9). Due to these pH excursions in the pore water, the solid-phase chemistry is also affected by the metabolic activity of the cable bacteria.

The acidification of the suboxic zone results in the dissolution of iron(II) sulfide (FeS) and calcium carbonate ($CaCO_3$), which causes accumulation of ferrous iron ($Fe^{2+}$), calcium ($Ca^{2+}$) and manganese ($Mn^{2+}$) in the pore water (Risgaard-Petersen et al., 2012; Rao et al., 2016; Sulu-Gambari et al., 2016; van de Velde et al., 2016). Upon release, these cations diffuse both upward

and downward in the sediment column (Fig. 1). The downward diffusion of $Fe^{2+}$ results in the precipitation of FeS once the ferrous iron encounters the sulfide appearance depth. The upward diffusion of $Fe^{2+}$ results in the formation of iron (hydr)oxides (FeOOH) once $Fe^{2+}$ reaches the oxic zone (Risgaard-Petersen et al., 2012; Rao et al., 2016). In a similar manner, $Ca^{2+}$ can diffuse upwards and downwards into sediment layers with a higher pH (and carbonate ion concentrations) and reprecipitate as $CaCO_3$, likely in the form of high-magnesium calcite (Risgaard-Petersen et al., 2012). This sometimes gives rise to a hard

$CaCO_3$-rich crust that rests on top of the sediment (Risgaard-Petersen et al., 2012; Rao et al., 2016).





Clearly, the presence and activity of cable bacteria has a large impact on the mineralogy of aquatic sediments, and results in the de novo formation of solid phases in the surface layer, like FeOOH, FeS and $CaCO_3$. Mineral formation can occur over a relative short time span. For example, Seitaj et al. (2015) showed that over the time course of a few weeks to months about 1

mole of FeS per $m^2$ is converted to FeOOH under the influence of cable bacteria. The exact mineralogy of these newly reprecipitated iron and carbonate phases is presently not known. It is also unknown to which degree the cable bacteria are responsible for the precipitation of these newly formed minerals. The processes in which microbes mediate mineral precipitation can be grouped into two modes: (1) biologically induced mineralization (BIM) and (2) biologically controlled mineralization (BCM) (Lowenstam and Weiner, 1989). Minerals that form by BIM generally nucleate and grow extracellularly

as a result of the metabolic activity of the organism and subsequent chemical reactions involving metabolic by-products. BIM is an unintended and uncontrolled consequence of metabolic activity. The formed minerals are generally characterized by poor crystallinity, broad particle-size distributions, and lack of specific crystal morphology. In essence, BIM is equivalent to inorganic mineralization under the same environmental conditions and the precipitated minerals are indistinguishable from minerals produced by inorganic processes. BIM is particularly significant for bacteria that respire sulfate and/or metal

(hydr)oxides where the metabolic products are reduced metal ions and sulfide, which are reactive and participate in subsequent mineral formation (Lowenstam and Weiner, 1989; Frankel and Bazylinski, 2003). BCM is a much more tightly regulated mineral formation process and the precipitated minerals have a physiological and structural role. The precipitation is regulated by the microbes in such a way that appropriate mineral saturation states are achieved and minerals can be formed within the organism even when conditions in the bulk solution are thermodynamically unfavorable. Minerals formed by BCM show a

high degree of crystallinity and have a specific crystal morphology (Konhauser and Riding, 2012).

Understanding bacterially mediated mineralization is crucial in understanding the complex interactions of biological, chemical and physical processes ( Konhauser, 1998a; Gonzalez-Munoz et al., 2010) since it affects the geochemical cycling of mineral-forming elements (e.g. C, Fe, S, Ca and O) on a short-term time scale up to geological time-scales (Konhauser and Riding,

2012). Moreover, little is known about how the cable bacteria themselves are affected by the biogeochemistry of their surroundings and how they maintain their population in an environment that is rapidly changing as a result of their own metabolic activity. Here, we present and discuss the results of an explorative study on cable bacteria using a combined approach of electron microscopic and spectroscopic techniques to characterize the different cell-mineral interactions.



## 2 Methods

### 2.1 Sediment collection and incubation

Enrichment cultures were initiated in which natural marine sediments where homogenized and incubated in the laboratory under conditions that stimulate the growth of cable bacteria (pure cultures of cable bacteria are not yet available). To this end, sediment was collected at four different locations, three of which are located in the Netherlands, and are referred to as Rattekaai Salt Marsh (RSM), Mokbaai (MB) and Marine Lake Grevelingen (MLG), while the fourth site is located in the Black Sea (BS). The RSM site (51.4391°N, 4.1697°E) is located within the creek bed of an intertidal salt marsh within the Eastern Scheldt tidal inlet. The MB site (53.00°N, 4.7667°E) is located at an intertidal flat near the island of Texel (Wadden Sea). Samples from MLG (51.4624°N, 3.5616°E ) were taken at a depth of 34 m. MLG is a seasonal hypoxic marine lake in the Dutch Delta area which experiences bottom water hypoxia and anoxia in the summer in its deep basins. The BS sediment samples were taken at a water depth of 27 m on the continental shelf of the Black Sea (44.5917°N; 29.1897°E). Sediments from RSM, MB and MLG were used because abundant cable bacteria populations have been previously documented under field conditions at the sites and cable bacteria enrichments were successfully obtained in previous experiments (Malkin et al., 2014; Burdorf et al., 2017). Surface sediments from RSM and MB were collected during low tide by collecting the first 5 centimeter of sediment with a shovel. For the collection of MLG and BS sediments, a gravity corer was used on-board ship. In all cases, the thin upper layer of oxidized sediment was discarded and the reduced, sulfidic sediment was homogenized and sieved to remove fauna (0.5 mm mesh size for RSM, MB and MLG, 4 mm mesh size for BS) before being repacked in plastic cores as described in Burdorf et al. (2017). Sediment cores were incubated in a temperature-controlled room (RSM and BS at 20 °C; MB and MLG at 15 °C) in the dark, submerged in artificial seawater at in situ salinity (RSM and MB 35; MLG 32; BS 17.9), and constantly aerated through air bubbling.

### 2.2 Filament extraction procedure

Individual cable bacterium filaments were retrieved from the incubated sediment enrichments for subsequent microscopic inspection. A number of different filament extraction protocols (FEP) were used, depending on the sediment and type of microscopy that was implemented. An overview of which preparation method was used for which sample is given in Supplementary Table 1.

FEP1: Filaments were picked from the incubated sediment cores under a stereomicroscope using custom-made fine glass hooks. Retrieved filaments or clumps of filaments were subjected to sequential washes by transferring them between separate droplet solutions on a microscope slide as described in Vasquez-Cardenas et al. (2015). Filaments were washed first three times in artificial seawater to remove sediment particles, then followed by several washes (> 3) in Milli-Q (Millipore, The Netherlands). Milli-Q washes were implemented to prevent precipitation of salt crystals during sample drying. Isolated filaments were subsequently pipetted onto polycarbonate filters (pore size 0.2 μm, Isopore, Millipore, Netherlands) that were



mounted on aluminum SEM stubs (1 cm diameter) fitted with conductive carbon tape and allowed to air-dry in a desiccator. After drying, samples were coated with a ~15 nm gold layer for SEM analysis or coated with a ~5 nm carbon layer for EPMA.

FEP2: A small amount of sediment was taken from the oxic top layer of the sediment core and transferred to a 15 mL Greiner tube which was then filled up to a volume of 10 mL with Milli-Q water. The Greiner tubes were centrifuged at 2,100 g for 2

min. This washing step was repeated three times. Samples were then transferred onto an SEM stub. This sediment was air-dried overnight before gold coating.

FEP3: Sediment containing cable bacteria was retrieved from the oxic top layer with a scalpel and directly transferred onto an SEM stub. This sediment was air-dried overnight before gold coating.

FEP4: Individual filaments were hand-picked from the sediment with custom-made glass hooks as in FEP1 and then washed

three times with artificial seawater followed by several (>3 washes) in Milli-Q. To enable for long-term storage, these samples were subsequently transferred to a 50% ethanol solution and stored in the freezer at -20ºC. Upon analysis, they were defrosted and pipetted onto a SEM stub with carbon conductive tape and air-dried before gold coating.

### 2.3 Microscopy and spectroscopy

Different microscopy and spectroscopy techniques were used to obtain structural as well as compositional information on the

minerals that were formed. An overview of the imaging modes used for the different samples is provided in Supplementary Table 1.

### 2.3.1 Epi-fluorescence microscopy

Epi-fluorescence microscopy provides information on the location and size of intracellular granules. Filaments extracted from the sediment were stained with the general DNA stain 4',6-diamidino-2-phenylindole (DAPI) and were imaged using a Zeiss

Axiovert 200M epifluorescence microscope (Carl Zeiss, Göttingen, Germany) equipped with the Zeiss filter set 02 (excitation G365, BS395; emission LP420). An excitation wavelength of 358 nm was used and emission was detected at 463 nm.

### 2.3.2 Digital Holographic Microscopy (DHM)

Digital holographic microscopy is a technique which not only allows the light intensity information of the investigated object

to be captured but also the so-called phase information in a quantitative way (Dubois et al., 2004). This allows for the imaging of transparent substances such as mucoids, biofilms and other types of extracellular polymeric substances (EPS) that are typically not visible in the light intensity image (Zetsche et al., 2016a, 2016b) or with SEM imaging.

Specimen for DHM were obtained by inserting glass slides into an enrichment culture from MLG and leaving the glass slide in the sediment for a period of days. Afterwards the slide was retrieved and washed with MilliQ. Some filaments remained

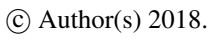



attached to the slide. Holograms of cable bacteria were captured with an oLine D$^3$HM digital holographic microscope (Ovizio Imaging Systems NV/SA, Belgium) and subsequently the respective light intensity and phase images derived as explained in more detail in Zetsche et al. (2016a).

### 2.3.3 Scanning electron microscopy (SEM) coupled to Electron dispersive X-ray spectroscopy (EDS)

SEM imaging was used to obtain high-resolution images of the cable bacterium filaments and their associated mineralogical
structures. SEM imaging was combined with electron dispersive X-ray spectroscopy (EDS) to obtain the elemental composition of the cable bacteria and of the related mineral structures at specific locations (called 'spots'). EDS analysis was also used to determine which elements should be further analyzed by X-ray element mapping. SEM-EDS analysis was performed on a Phenom ProX desktop SEM (Phenom-World B.V., the Netherlands). SEM images were obtained under a 0.1-0.3 mbar vacuum and a high accelerating voltage (10 kV or 15 kV). Because of the small size of the cable bacteria, the
interaction volume of the EDS analysis was assessed by Monte Carlo simulations with Casino 2.48 software (Drouin et al., 2007) to check if the measured X-rays are representative of the chosen spot.

### 2.3.4 Electron Probe Micro Analysis (EPMA)

EPMA imaging was combined with X-ray element mapping using wavelength dispersive X-ray spectroscopy (WDS) as well as EDS. EPMA coupled to WDS provides a much higher spectral (2-15 eV) resolution than EDS, which has a spectral
resolution between 70 and 180 eV. Therefore, WDS X-ray mapping allows for more accurate element maps (Goldstein et al., 2003). The main advantage of EDS mapping is the capacity to collect data in parallel for the entire range of photon energies excited by the primary beam. This allows all elements to be mapped in a single scan. In contrast, WDS is restricted to a more narrow energy window and therefore only one element can be mapped in a single scan (Goldstein et al., 2003).

EPMA imaging and the associated X-Ray element mapping (WDS and EDS) was performed on a JXA-8530F Hyperprobe
Field Emission Electron probe micro-analyzer outfitted with five automated wavelength dispersive X-Ray spectrometers (JEOL, Japan). This electron microprobe was equipped with five WDS units that allowed for the simultaneous collection of five element maps during a single run. EPMA analysis and element mapping was performed under high vacuum ($10^{-5}$-$10^{-6}$ mbar), high voltage (15 kV) and a probe current of 10 nA. X-ray element maps of phosphorus, calcium, magnesium, iron and sulfur were obtained using WDS. X-ray element maps of silicon and aluminum were collected using EDS. Pixel sizes were
0.30 µm and dwell times at each pixel were 50 ms or 240 ms. Longer dwell times resulted in beam-damaged sample erosion. SEM images prepared from the backscattered electron signal in the JEOL electron probe micro-analyzer were collected under high vacuum and high accelerating voltage (15 kV).



### 2.3.5 Focused Ion Beam – Scanning Electron Microscopy (FIB-SEM)

Specimen for FIB-SEM were obtained by inserting ACLAR 7.8 Mil slides (Electron Microscopy Services) into an enrichment
culture and leaving the ACLAR in the sediment for a period of days. Afterwards the ACLAR slide was retrieved and washed
with MilliQ. Some filaments retained firmly attached (cemented) to the ACLAR slide. These ACLAR slides with filaments
were incubated in freshly prepared fixative (2% paraformaldehyde (PFA, Applichem), 2.5% gluteraldehyde (GA, EMS) in
0.15M sodium cacodylate (Sigma-Aldrich) buffer, pH 7.4) at room temperature (RT) for 30 min. Fixative was removed by
washing 5 x 3 min in 0.15M cacodylate buffer and samples were incubated in 1% osmium ($OsO_4$, EMS), 1.5% potassium
ferrocyanide (Sigma-Aldrich) in 0.15M cacodylate buffer for 40 min at RT. This was immediately followed by a second
incubation in OsO4 (1% $OsO_4$ in double-distilled (dd)$H_2O$) for 40 min at RT. After washing in dd$H_2O$ for 5 x 3 min, samples
were incubated overnight at 4ºC in 1% uranyl acetate (UA, EMS). The next day, UA was removed by washing in dd$H_2O$ for
5 x 3 min. After final washing steps the samples were dehydrated using ice-cold solutions of increasing EtOH concentration
(30%, 50%, 70%, 90%, 2x 100%), for 3 min each. Subsequent infiltration with resin (Durcupan, EMS) was done by first
incubating in 50% resin in EtOH for 2 h, followed by at least 3 changes of fresh 100% resin (including 1 overnight incubation).
Next, samples were embedded in fresh resin and cured in the oven at 65°C for 72 h. Before FIB-SEM imaging, ACLAR was
removed from the polymerized resin, leaving the filaments embedded directly at the surface, the resin block was mounted on
aluminum SEM stubs and samples were coated with ~6 nm of platinum (Quorum Q150T ES). FIB-SEM imaging was
performed using a Zeiss Auriga Crossbeam system. The Focused Ion Beam (FIB) was set to remove 10 nm sections by
propelling gallium ions at the surface. In between the milling of the sections, samples were imaged at 1.5kV using an ESB
(back-scattered electron) detector.

## 3 Results

A wide number of samples (>50 samples, >500 images) from enrichment cultures with cable bacteria activity were
microscopically screened for minerals that were directly associated with the cable bacterium filaments, i.e., they occurred
inside or on the outer surface of the filaments. SEM imaging revealed three types of mineral transformation closely associated
with cable bacteria (Fig. 2). (1) The intracellular formation of ***polyphosphate granules***, (2) the ***coating*** of filaments with
existing mineral particles, in particular of clay particles, and (3) the ***encrustation*** of the cable bacteria filaments with newly
deposited minerals, which is observed as an electron dense layer surrounding the surface of the filaments.

In several samples from MB and RSM, cable bacteria without of any form of precipitation, as well as cable bacteria with poly-
P granules, coating or encrustation, were all present in the same sample, thus suggesting heterogeneous precipitation patterns
(Fig. 5b and 10).



### 3.1 Intracellular polyphosphate granules

Naked filaments are defined as filaments with no visible extracellular attachment of mineral particles and no visible encrustation. These naked filaments comprised the majority of filaments encountered in the sediment samples, and were

frequently observed to contain circular to ellipsoidal granules when imaged with SEM or when using fluorescence microscopy with DAPI staining. Note that coated or encrusted filaments may also contain these granules, but this could not be verified, as the presence of the intracellular granules is obscured by the surrounding sheath or encrustation. DAPI showed the granules as bright blue spots in the cells of naked filaments (Fig. 3a), thus suggesting they are polyphosphate (poly-P) inclusions. DAPI is a fluorescent dye usually used for DNA staining but when DAPI is used at a high concentration, it also stains polyphosphate

granules (Streichan et al., 1990)

SEM imaging provided higher resolution images, which depicted the granules as electron-dense, bright, white spots (Fig. 3b-c). There was considerable variation in the size and amount of granules present between filaments; some filaments contained no granules, others contained a single granule or a few large ones, while still other filaments contained multiple smaller granules dispersed throughout the cell. There was also variation in poly-P patterns between cells of the same filament, though

typically, the granule pattern remained similar between neighboring cells of the same filament (Fig. 3c).

SEM-EDS spot analysis on the inclusions indicated large peaks of phosphorus (P) and oxygen (O) in the resultant spectrum, accompanied by smaller but prominent peaks of calcium (Ca), magnesium (Mg) and also sulfur (S) (Fig. 3d). Qualitative chemical analysis using EPMA-WDS confirmed that the poly-P granules within non-encrusted filaments contained P, Ca and Mg, and that P was correlated with both Ca and Mg (Fig. 4). The spatial co-localization of P with Ca, Mg, S and Fe was

evaluated using Pearson's correlation coefficient for two different WDS X-Ray element maps of naked filaments (Fig. 4, Supplementary Fig. S1). The respective correlation coefficients varied from 0.76 to 0.94 for Ca, between 0.64-0.83 for Mg, 0.55-0.59 for S and 0.12-0.2 for Fe. Accordingly, there is a significant, high correlation between P and Ca and between P and Mg, and a moderate correlation between P and S throughout the filament. The low correlation between P and Fe suggests that the observed correlations between P and Ca and P and Mg are not because of a confounding factor (e.g. dilution by a common

element or the structure of the cell), because then there would have been a higher correlation between the P and Fe as well.

### 3.2 External coating of filament

Sporadically, SEM imaging revealed that parts of a cable bacterium filament were covered by a heterogeneous coating of mineral particles that appeared to be attached to the surface of the filaments (Fig. 2b, 5). The shape and size of the incorporated minerals varied, which provides the coating with a rough and rugged texture. DHM imaging showed the presence of living

cable bacteria coated with these particles (Fig. 7a-b) as well as a remaining sheath of extracellular polymeric substances (EPS) (Fig. 7c). Filaments covered with a particle coating were generally extracted together with naked filaments that had poly-P granules from the same sediment batch (Fig. 6a). WDS and EDS element mapping revealed that the attached mineral particles



predominantly consisted of Si, Al, Mg and Fe (Fig. 6 c-f), and therefore, they are most likely ambient clay-like particles. Coated filaments showed a clearly different signature in the WDS and EDS element mapping compared to naked filaments.

The latter were clearly identifiable in the P map (Fig. 6b), but did not show any signature for Si, Al, Mg and Fe (Fig. 6c-f). Both coated filaments and naked filaments contain P throughout the filament, although the presence of P was less prominent in the coated filaments (Fig. 6b), suggesting a higher concentration of P in naked filaments and an inverse relationship between external clay coating and poly-P formation. However, most of the X-ray counts measured came from the sample surface and therefore WDS element maps reflect the composition of cable bacteria living in the oxic zone of surface sediments. Since the

surface of the coated filaments contains minerals, the P content of the filaments from different fields of view cannot be compared using WDS element maps.

### 3.3 Filament encrustation

Filament encrustation involved the deposition of a solid mineral phase along the external surface of cable bacterium filaments, and this solid mineral crust was tens or even hundreds of nanometers thick (Fig. 8). Filament encrustation was observed quite

regularly in sediment samples, but importantly, it was only present in filaments that were retrieved from the oxic zone of the sediment enrichments.

Encrusted filaments showed a distinct morphology when compared to the naked or coated filaments. Based on the morphology, different degrees of encrustation were distinguished. If a filament was weakly encrusted it closely followed the topography of the outer cell surface of the cable bacterium filaments, as if a homogeneous layer of mineral had precipitated on the outer

surface (Fig. 8). Cable bacteria have a characteristic outer surface morphology, which consists of a pattern of parallel ridges that run all along the length of a filament (Pfeffer et al., 2012). This longitudinal ridge pattern was often still clearly visible in the encrusted filaments, even if filaments showed a high degree of encrustation (Fig. 8a-f). Filaments that were broken at cell junctions provided a cross-sectional view of the encrustation, and this allowed to estimate the thickness of the encrusted layer, which may reach up to 430 nm (Fig. 8a). In filaments that were only lightly encrusted, the individual encrusted ridges were

separately visible (Fig. 8b) and the cell junction was clearly identifiable as a ring containing apparent voids in encrustation (Fig. 8a-b). The presence of these voids at the cell junctions fully aligns with the model for the internal structure of the cable bacteria as recently proposed by Cornelissen et al. (subm.). This model proposes that the outer cell membrane shows invaginations around fibers that run from cell to cell, thus creating a ring of regularly spaced voids at the cell junctions. When the encrustation becomes thicker (Fig. 8c-f), the ridge pattern and the voids at the cell junctions become less apparent as these

features are gradually obscured by the deposition of new layers of mineral.

FIB-SEM analysis was done on filaments that were weakly encrusted (Video 1, Supplementary information) as well as on filaments that were strongly encrusted (Video 2, Supplementary information). The mineral that formed the crust was strongly stained with the uranyl acetate and/or osmium tetroxide stains that were used in sample preparation for FIB-SEM, thus providing a dark seal around the filaments in the FIB-SEM images (Fig. 9). In contrast, the ridge compartments themselves





were left unstained, and hence appeared white in the FIB-SEM images. This allowed us to estimate the thickness of the crust by measuring the distance from the top of the ridge compartment to the top of the crust. The weakly encrusted filaments all had 16 ridge compartments in the cell envelope, while the strongly encrusted filaments had 11 ridge compartments. Both sets of filaments were obtained from the same sample, and hence, this confirms that cable bacteria within the same environment are highly variable with respect to their number of ridges, as recently put forward by Cornelissen et al. (subm). We estimated that on average 180 nm of mineral was deposited on top of the weakly encrusted filaments. In contrast, the crust of the strongly encrusted filaments was much thicker and amounted to 500 nm. These latter filaments were so highly encrusted that crusts from different filaments were cemented together by the mineral precipitation and the thickness along a filament was more heterogeneous when compared to the weakly encrusted filament (Video 2, Supplementary information).

EDS spot analysis (Fig. 8f) and WDS analysis (Fig. 10 and 11) revealed that the encrustations were mainly composed of O, Fe and P. Minor amounts of Ca and Si were sometimes also observed, but these did not consistently show up among samples, indicating that the Ca and Si content of the crusts was variable. This EDS spot analysis was repeated on multiple filaments (~50) of different samples and consistently showed the same elemental composition (presence of O, Fe, P and sometimes Si, Ca and S) in the mineral encrustation (data not shown).

A second type of encrustation morphology shows nanometer-sized globules present on the surface of cable bacteria filaments (Fig 8a and 8c). WDS analysis showed that these globules also contained Fe and P (Fig. 11). Some filaments covered in these globules were less mineralized than the completely encrusted cable bacteria suggesting that the growth of these iron-phosphate minerals predates the more advanced crust formation (Fig. 11). Some filaments were completely encrusted and were also covered in globules showing that they continued growing after the cell is encrusted (Fig. 8a and 8d).

## 4 Discussion

In the present study, we examined enrichment cultures of cable bacteria by different types of microscopy and spectroscopy, and observed three different types of mineral formation directly associated of with multi-cellular filaments of these cable bacteria: the formation of poly-P granules within the cells, the attachment of clay-particles into a coating surrounding the bacteria, and encrustation of the cell envelope by iron minerals.

### 4. 1 Polyphosphate (Poly-P) granules

In the laboratory sediment enrichments from all field sites examined in this study, naked cable bacterium filaments were observed that contained poly-P granules. It was not possible to discern if these poly-P granules were also present in encrusted or coated cable bacteria. Both the size and number of poly-P granules varied widely between filaments present in the same sediment sample (Fig. 3a-c). Some filaments contained no poly-P granules, other filaments had one large poly-P granule in each cell, while still other filaments contained multiple smaller poly-P granules in each cell. Differences in size and number



of poly-P granules could even be observed within a single bacterium (Fig. 3b,d, Fig. 4). Yet in general, the differences in size and number of the poly-P granules were larger between filaments than within cells belonging to the same filament.

The EDS and WDS spectra showed that granules mainly contained P and O, which suggested they consist of poly-P granules, which are long-chain polymers consisting of 2-1000 orthophosphate residues linked together by a high energy phosphoanhydride bond that is similar to the bond in ATP (Kornberg, 1995; Seufferheld et al., 2008; Rao et al., 2009). Poly-

P appears to have distinctive biological functions in microbial cells, and this function depends on abundance, chain length, origin, and subcellular location of the granules. Poly-P granules have been thought to act as an ATP substitute and energy storage, although the metabolic turnover of ATP is considerably higher than that of poly-P (Kornberg, 1995). They can also be a reservoir for orthophosphate ($P_i$), a chelator of metal ions, and a buffer against alkali ions. Finally, poly-P granules have been claimed to aid the channeling of DNA, and to regulate the responses to stresses and adjustments for survival, especially

in the stationary phase of culture growth and development (Kornberg 1995 and references therein). Due to their anionic nature, poly-P granules typically form complexes with cations (Kornberg, 1995; Rao et al., 2009; Seufferheld et al., 2008). This is fully confirmed by our EDS measurements, which show prominent peaks of Ca and Mg, as well as by the X-Ray mapping, which revealed a high spatial correlation between P on the one hand, and Ca and Mg on the other hand. Moreover, this correlation pattern was consistent across filaments with different numbers and sizes of granules (Fig. 4, Supplementary Fig.

S1).

Our EDS data also showed the presence of S in the poly-P granules (Fig. 3d). However, X-Ray mapping illustrated that S showed a poor spatial correlation with P (Fig. 4, Supplementary Figure S1). Hence, sulfur appeared not restricted to the poly-P granule, but was evenly distributed throughout the entire filament (Fig. 4, Supplementary Figure S1). Given this homogeneous distribution, we hypothesize that sulfur is not linked to poly-P granules, but is an abundant element in the thick

cell envelope that surrounds the cable bacterium filaments (Jiang et al., 2018; Cornelissen et al., subm).

Poly-P granules have been observed in other large sulfur-oxidizing bacteria, like *Thiomargarita, Thioploca* and *Beggiatoa* (Schulz and Schulz, 2005) as well as in cable bacteria under *in situ* conditions in a coastal hypoxic basin (Sulu-Gambari et al., 2016). Both the size and number of poly-P granules are thought to greatly depend on the environmental conditions these large sulfur-oxidizing bacteria are exposed to, such as the sulfide and oxygen regime (Brock and Schulz-Vogt, 2011). Under oxic

conditions, *Thiomargarita namibiensis* and *Beggiatoa* gain energy from aerobic oxidation of elemental sulfur that is intracellularly stored in S globules. The excess energy gained through sulfur oxidation (i.e. the energy that does not need to be invested in overall metabolism and cell growth) then initiates luxury uptake of P, which is stored in poly-P granules. Under anoxic conditions, *T. namibiensis* and *Beggiatoa* gain energy from the oxidation of free sulfide ($H_2S$) to elemental sulfur using internally stored nitrate as an electron acceptor. This oxidation does not yield as much energy and so the breakdown of poly-

P then forms an auxiliary mechanism to gain energy (Brock and Schulz-Vogt, 2011; Schulz and Schulz, 2005).



Acidocalcisomes are subcellular organelles in eukaryotes that are homologous to Poly-P granules in prokaryotes (Seufferheld et al., 2008). Acidocalcisomes are an electron-dense, acidic compartment containing a matrix of pyrophosphate and polyphosphates with bound calcium and other cations, mainly magnesium and potassium. Acidocalcisomes possess an enclosing membrane that has a function in the calcium and pH homeostasis (Seufferheld et al. 2003), as well as the osmotic

homeostasis (Docampo et al., 2005). It has been found that poly-P granules in the bacteria *Agrobacterium tumefaciens* (Seufferheld et al., 2003) and *Rhodospirillum rubrumare* (Seufferheld et al., 2004) are similar to the acidocalcisomes found in eukaryotic species; they are rich in orthophosphate ($P_i$), pyrophosphate ($PP_i$), poly-P, Ca, Mg and potassium (K), and are acidic and enclosed by a membrane. In addition, Seufferheld et al. (2003, 2004) discovered that the acidocalcisomes in eukaryotic species as well as the poly-P granules in *A. tumefaciens* and *R. rubrumare* both contained the enzyme protonpyrophosphatase

($H^+$-PPase), which suggests a common origin of these organelles. The storage of P and the complexed cations as poly-P reduces the osmotic effect of large pools of the compounds (Docampo and Moreno, 2012). The Ca in the acidocalcisome is bound to a polyanionic matrix of poly-P, and can be released after alkalinization of the organelle. $Ca^{2+}$ uptake into acidocalcisomes is driven by $Ca^{2+}$-ATPases in several protists and probably through $Ca^{2+}/H^+$ antiporters facilitated by the proton pumps in other organisms (Docampo and Moreno, 2012). Recently, poly-P granules that are similar to acidocalcisomes have also been

observed in the sulfide-oxidizing marine *Beggiatoa* strain 35Flor (Brock et al., 2012). These poly-P granules showed a similar association between P, Ca and Mg and were enclosed by a lipid layer which is hypothesized to be a membrane. However, in contrast to acidocalcisomes, the poly-P granules found in strain 35Flor are not acidic (Brock et al., 2012).

Given the correlation between P, Ca and Mg as seen in the WDS mapping (Fig. 4) and EDS analysis (Fig. 3d), we hypothesize that the poly-P granules seen in the cable bacteria could be membrane-bound vacuoles that have a function in the $Ca^{2+}/H^+$

homeostasis. This could allow cable bacteria to maintain optimum intracellular pH levels in an environment with strong pH gradients, specifically in the alkaline oxic zone. Another possibility is that the poly-P granules function as a kind of 'energy safety system' that enables survival under conditions of low redox potential, similar to the function of the poly-P granules in other sulfur-oxidizing bacteria such as *Beggiatoa*, *Thioploca* and *Thiomargarita* (Brock and Schulz-Vogt, 2011). One option could be that cable bacteria use the Poly-P granules as an internal energy reservoir that drives their motility (Bjerg et al., 2016)

under energy-starved conditions (e.g. when a filament is no longer in contact with the oxic zone, and hence cannot perform long-distance electron transport due to a lack of electron acceptor). To investigate if the poly-P granules within cable bacteria are separate organelles within the cells and if they function as an 'energy safety system' similar to poly-P granules in other sulfur-oxidizing bacteria, the presence of a membrane around the poly-P granules needs to be confirmed as well as the uptake of phosphate during oxic conditions and the release of phosphate during anoxic and sulfidic conditions (Brock et al., 2012).

This would provide insight into the ways cable bacteria maintain their function in an environment that is variable on both spatial and temporal scales as a result of their own metabolism.



### 4.2 External coating of mineral particles

In a number of cases the filaments of cable bacteria are surrounded by a heterogeneous coating in which mineral particles of different sizes and geometries are embedded. One option is that the attached minerals are formed de novo; another – and more likely option – is that existing minerals are glued together via extracellular polymeric substances (EPS). Cable bacteria show a gliding motility where filaments follow the receding sulfide front and stay connected with the oxygen whilst the gap between the oxygen and sulfide is widening as a consequence of the growth and metabolism of the cable bacteria (Bjerg et al., 2016). Once in contact with oxygen, the cable bacteria were observed to stop gliding. This behavior is suggestive of oxygen chemotaxis. As long as the oxygen front is changing, cable bacteria will glide through the sediment (Bjerg et al., 2016) most likely by excreting EPS. Digital holographic microscopy confirms the formation of EPS in relation to the gliding movement (Fig. 7). The DHM images show a sheath of optically thick EPS that tightly surrounds the filament. Moreover trails of EPS are left behind on aclar slides embedded in the sediment, suggests that EPS is produced by the filaments as an aid in motility (Fig 7). EPS excretion to aid motility has been found in other filamentous bacteria like cyanobacteria and *Beggiatoa* spp. (Larkin and Henk, 1996; Risser and Meeks, 2013). In *Beggiatoa* spp. pores in the cell wall have been observed, and the hypothesis is that EPS is excreted through these pores to sustain gliding motility (Larkin and Henk, 1996). In cyanobacteria, similar evidence for gliding motility by EPS excretion (like the presence of excretion pores and the staining of trails) is supported by the recent discovery of an essential EPS excretion gene (Risser and Meeks, 2013).

*Thioploca* spp. cells form filaments that cling to each other and secrete a sheath of mucous that surrounds several filaments. This sheath allows the filaments to tunnel through the sediment up to the overlying water (Larkin and Strohl, 1983). These *Thioploca* sheaths are often covered with detritus (Rickard, 2012). We propose that the gliding movement of cable bacteria might "trap" already present clay (nano)particles in the EPS matrix resulting in the extracellular attachment of clay particles without the formation of new clay minerals  (Fig. 2e). The interaction between particulate minerals and microbial cells may arise because of the binding of pre-formed, finely dispersed minerals, such as colloidal silica, metal oxides and clays (Phoenix et al., 2005). Whether the extracellular attachment has any (negative) effect on the metabolism of the cable bacteria is unknown and open for investigation.

### 4.3 Cell encrustation

The combination of SEM, EDS, EPMA, WDS and FIB-SEM analysis showed the presence of cable bacteria cells that were encrusted with a mineral layer containing mainly Fe and P and minor amounts of Mg, Ca, and Si (Fig. 7-10). The encrustation was only observed in (parts of) cable bacteria that were present in the oxic zone. We propose that the encrustation consists of amorphous or poorly-crystalline iron(oxyhydr)oxides that precipitated onto the outer membrane thereby either incorporating or adsorbing phosphate.




The external surface of a cable bacterium filament shows a unique structure with uniform ridges running along their entire length of the filament (Pfeffer et al., 2012; Meysman, 2018). The ridges contain fiber-like structures (Cornelissen et al., subm), and the hypothesis is that these fibers have the capacity to transport electrons along the filament (Meysman, 2017). The fibers

are running inside the continuous periplasmic space, with the collective outer membrane serving as electrical insulation from the external medium (Cornelissen et al . subm). Adjacent cells within the filaments are separated by cell junctions that are bridged by the periplasmic filling and enclosed by the collective outer membrane (Jiang et al., 2018; Cornelissen et al . subm). These ridge structures were also observed in the mineralized filaments (Fig. 7 and 8). Therefore, we refer to the formation of this mineral layer as ''cell encrustation'', which is a term borrowed from Miot et al. (2009), who found that the morphology of

the periplasm was preserved upon mineralization of nitrate reducing $Fe^{2+}$ oxidizing bacteria strain BoFeN1 (Miot et al., 2009, 2011). Even though the encrustation is not found within the periplasmic space (Fig. 8), it preserves the ridge structure that is associated with the fibers in the periplasm.

Cable bacteria belong to the gram-negative bacteria and there have been multiple reports on the encrustation of other gram-negative bacteria (Benzerara et al., 2004, 2008, 2011; Goulhen et al., 2006; Miot et al., 2009, 2011; Schädler et al., 2009),

which suggests that this might be a widespread and common process. These are all bacteria that apparently have no mechanism to avoid cellular encrustation. Several $Fe^{2+}$ oxidizing bacteria have evolved mechanisms to avoid cellular encrustation by $Fe^{3+}$ minerals; some species create more acidic micro-environments resulting in a higher solubility of $Fe^{3+}$ ions (Schädler et al., 2009), *Gallionella* spp. and *Leptothrix* spp. produce extracellular organic polymers that nucleate $Fe^{3+}$ precipitates leading to the formation of spirally twisted stalks and sheaths, respectively (Hallberg and Ferris, 2004). *Mariprofundus ferrooxydans*

strain PV-1 have hydrophilic cell surfaces and a near-neutral charge to prevent encrustation (Melton et al., 2014). Cable bacteria appear not to have developed a mechanism to avoid encrustation which might be problematic for their survival. Cell encrustation could potentially limit the diffusion of substrates and nutrients to the cell, impair uptake of these compounds across the membrane, and as a consequence lead to the stagnation of cell metabolism and eventually even to cell death (Konhauser, 1998b; Schädler et al., 2009).

Gram-negative bacteria typically possess an outer membrane that forms an asymmetric lipid-protein bilayer which incorporates phospholipids, lipopolysaccharides (LPS) and proteins, and separates the external environment from the periplasm. The outer face contains almost all of the LPS while the inner face holds most of the phospholipid. The LPS layer is mostly anionic in nature due to the presence of exposed phosphoryl and carboxyl groups that can be readily ionized. This way, the LPS heavily interacts with the surrounding environment (Beveridge, 1999) and can be a site for the precipitation of fine-grained minerals

(Fortin et al., 1997), such as $Fe^{3+}$ minerals. Although the bacterial surface has a net negative charge, positively charged free amine groups are also present within the outer cell membrane. These groups can interact with negatively charged silicate ions at circumneutral pH values. Even if there are not enough amine groups present, superfluous silica ($SiO_3^{2-}$) might be deposited through metal ion bridging where a multivalent metal ion (e.g. $Fe^{3+}$ or $Ca^{2+}$) cross-links $SiO_3^{2-}$ to carboxyl or phosphoryl groups via electrostatic interactions (Schultze-Lam et al., 1996). The binding of metal ions onto the cell membrane might be



dependent on a "proton motive force" (Mera et al., 1992). When protons are forced outside of the cell, which occurs within cells of the cable bacteria in the suboxic and sulfidic zone resulting in the lowered pH values that are part of the "geochemical fingerprint" (Meysman et al., 2015), the protons might preferentially bind to the cell membrane thereby influencing the metal-binding capacity. This gives rise to a microenvironment that is comparable to the macro-environment (lower pH values) stimulating dissolution of minerals and therefore inhibits the formation of an extracellular mineral crust in the more acidic

suboxic zone. In addition, the binding of protons to the outer membrane results in less binding of positively charged metal species such as $Fe^{2+}$ and $Fe^{3+}$.

The exact mineralogy of the iron oxides in the crust is unknown but the orange color suggests that it is mainly composed of amorphous or poorly-crystalline $Fe^{3+}$ minerals (e.g. ferrihydrite). The biogeochemical cycles of iron and phosphorus are intimately linked and phosphate ($PO_4^{3-}$) adsorption to, or co-precipitation with, iron (oxyhydr)oxides is an important

mechanism for P removal in both marine and freshwater environments. Research in Lake Grevelingen (The Netherlands) showed that the presence of cable bacteria promoted the formation of iron oxides and the removal of pore water P that was sequestered in the sediment as iron (oxyhydr)oxide-bound P (Seitaj et al., 2015; Sulu-Gambari et al., 2016). Due to the charged outer surface of the bacterial membrane, sorption of positively charged metal ions ( Ferris et al., 1987; Fortin et al., 1997; Beveridge, 1999; ), such as $Mg^{2+}$ and $Ca^{2+}$, as well as negatively charged silicate ions via cross-linking (Schultze-Lam et al.,

1996), might have resulted in the incorporation of minor amounts of Mg, Ca, and Si. The exact mineralogy and crystallinity of the minerals formed onto the cable bacteria and the interaction with organic groups remain unknown and can only be resolved with high-resolution synchrotron-based microscopy techniques such as scanning transmission X-Ray microscopy (STXM) (Miot et al., 2014 and references therein).

Iron (oxyhydr)oxides are widespread and form in any environment where $Fe^{2+}$-bearing waters come into contact with $O_2$

(Konhauser and Riding, 2012). In electrogenic sediments, the formation of an iron oxide crust has been observed in both laboratory experiments (Risgaard-Petersen et al., 2012; Rao et al., 2016;) as well as in-situ (Seitaj et al., 2015; Sulu-Gambari et al., 2016). It is an example of biologically induced mineralization (Lowenstam and Weiner, 1989) resulting from the metabolic activity of the cable bacteria and the subsequent availability and re-oxidation of the $Fe^{2+}$ ions in the oxic zone. The oxidation of $Fe^{2+}$ could be either abiotic or biotic. The abiotic oxidation rate of $Fe^{2+}$ by molecular oxygen is very slow at acidic

pH but increases steeply up to a pH of ~8. At neutral pH, the abiotic oxidation of $Fe^{2+}$ occurs within minutes (Stumm and Morgan, 1996). At pH values above 8, the oxidation rate is fast, but no longer varies with the pH. The rate of oxidation is both thermodynamically and kinetically enhanced by adsorption of dissolved iron species to hydrous oxide surfaces (Morgan and Lahav, 2007). For the $Fe^{2+}$ oxidation to be biotic, $Fe^{2+}$ oxidizing bacteria need to outcompete the abiotic reaction to have viable living circumstances. The twisted stalks of the $Fe^{2+}$ oxidizing *Gallionella* spp. have been found in samples of encrusted cable

bacteria (Fig. 8d) showing that, despite the high pH values, $Fe^{2+}$ oxidizing bacteria (partly) outcompete the abiotic reaction. There is also direct evidence for the co-existence of active cable bacteria and $Fe^{2+}$ oxidizing and $Fe^{3+}$ reducing bacteria in sediments representative of typical marine environments (Otte et al., 2018). Whenever cable bacteria were abundantly present



(0.1%-4.5%), both $Fe^{2+}$ oxidizing and $Fe^{3+}$ reducing bacteria were homogeneously distributed throughout the sediment and their presence was therefore decoupled from the traditional geochemical gradients (Otte et al., 2018). The presence of $Fe^{2+}$

oxidizing bacteria in sediments dominated by cable bacteria suggest a cooperation between these bacteria where the electrons generated by $Fe^{2+}$ oxidation are transferred to the cable bacteria. After oxidation, ferric iron (hydr)oxides are expected to precipitate more or less instantly at neutral or alkaline pH values due to the low solubility of $Fe^{3+}$ under these conditions. In many cases this occurs directly where the $Fe^{3+}$ is formed. If present in the proximity of cells, $Fe^{3+}$ ions, $Fe^{3+}$ complexes, $Fe^{3+}$ colloids and $Fe^{3+}$ minerals would therefore be expected to adsorb to prokaryotic cell surfaces that are generally effective

sorption interfaces for metal ions (Beveridge, 1999; Ferris et al., 1987; Fortin et al., 1997) as well as negatively charged silicate ions (Schultze-Lam et al., 1996).

In addition to the cellular encrustation that pertains to the length of the ridges, another form of mineralization in the form of nano-sized globules was observed (Figs. 5b, 8c, 10). Research on the biomineralization of strain BoFeN1 showed similar precipitation of iron phosphate globules at the cell surface (Miot et al., 2009). Those globules were observed before and after

complete cellular encrustation although they were more numerous and thicker with increasing degrees of cellular encrustation (Miot et al., 2009). Laboratory experiments on the interaction between bacterial surfaces and mineral particles with *Shewanella putrefaciens*, a gram-negative dissimilatory metal-reducing bacteria, showed that nano-sized particulate iron oxides could adsorb irreversible to the bacterial surface suggesting that not all cell surface-associated minerals nucleated on the cell envelope (Glasauer et al., 2001). On the basis of the available data either scenario is possible; (i) the globules were formed either while

the cable bacteria became encrusted by simultaneously nucleating onto the cell wall and continued growing during and after cellular encrustation, or, (ii) the globules were pre-formed nanominerals that irreversibly attached to the outer membrane and continued to grow after adsorption.

It appears that the metabolism of cable bacteria results in a cascade of reactions that eventually results in the inadvertent mineralization of their cells present in the oxic zone where the negatively charged phosphoryl and carboxyl surface groups on

their outer membrane are a site for mineral nucleation. Since the mineral precipitation does not appear to be controlled by the cable bacteria it is an example of biologically induced mineralization. The formation of a mineral crust on a bacterium's cell surface would potentially limit their cell metabolism and may eventually lead to cell death (Konhauser, 1998a; Schädler et al., 2009). The possible effect on cable bacteria metabolism of the encrustation is unknown and needs to be resolved.

**5 Conclusions**

Combined electron microscopy and spectroscopy on cable bacteria showed different types of mineral formation based on the morphology, location and chemical composition of the minerals: poly-P granules, extracellular coating by clay minerals and cellular encrustation by iron(oxyhydr)oxides in the oxic zone. The co-localization of Ca and Mg within the poly-P granules and the presence of Fe and P within the cellular encrustation was confirmed. Although the encrustation develops over time,

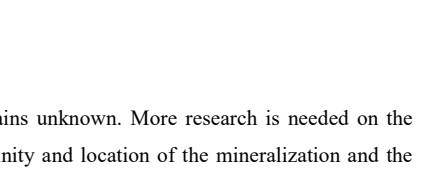

the effect of the encrustation on the metabolism of the cable bacteria remains unknown. More research is needed on the
succession of mineralization, the exact morphology, mineral phase, crystallinity and location of the mineralization and the
effect of mineralization on the metabolism of the cable bacteria. The cellular encrustation shows similarities with other
encrusting gram-negative bacteria (Benzerara et al., 2004, 2008, 2011; Goulhen et al., 2006; Miot et al., 2009, 2011; Schädler
et al., 2009), indicating a possible common mechanism for the mineralization of gram-negative bacteria that have no
mechanism to avoid encrustation.

**Author contributions**

NMJG, EMZ and FJRM designed the research. NMJG, EMZ, SHM and FJRM performed the analyses. NMJG, EMZ and
FJRM interpreted the data. NMJG wrote the paper with contributions provided by EMZ, JJM and FJRM.

**Acknowledgements**

We thank Laurine Burdorf for her help with the SEM imaging, Tilly Bouten and Segui Matveev for their guidance and help
with EPMA analysis, Anton Tramper for monitoring the incubations and Martijn Hermans for providing us with the samples
from the Black Sea. NMJG is the recipient of a Ph.D. scholarship for teachers from NWO in the Netherlands (grant
023.005.049). FJRM was financially supported by the European Research Council under the European Union's Seventh
Framework Programme (FP/2007-2013) through ERC Grant 306933, by the Research Foundation Flanders via FWO grant
G031416N, and the Netherlands Organization for Scientific Research (VICI grant 016.VICI.170.072). JM was supported by
the Ministry of Education via the Netherlands Earth System Science Centre (NESSC). The Hyperprobe Field Emission micro-
analyzer was partly supported by a NWO large infrastructure subsidy to JJM (175.010.2009.011).

**Data availability**

Video data (Video 1 and Video 2) will be made available at https://av.tib.eu/ after publication and will receive a separate
DOI.


**Competing interest**

The authors declare that they have no conflict of interest.





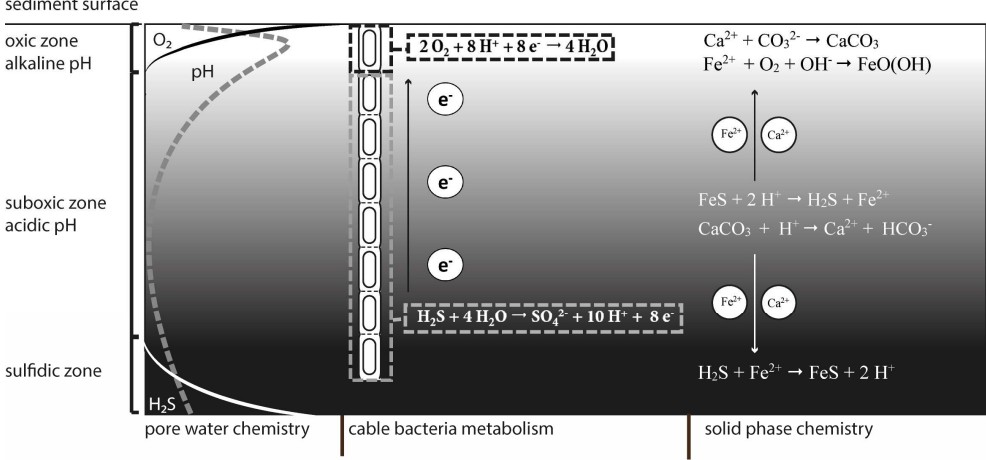

**Figure 1:** Microbially and chemically mediated reactions involved in the iron, sulfur and calcium cycle in electrogenic sediments. The electrogenic sulfur oxidation executed by cable bacteria spatially separates the redox half-reactions; the oxidation of sulfide takes place in the suboxic zone and generates electrons ($e^-$). These electrons are then transported through the cable bacteria and used via oxygen reduction. The sulfide oxidation reaction results in proton release and a decrease in pH while the oxygen reduction requires protons, which increases the pH of the pore-water in the oxic zone. The decreased pH in the suboxic zone promotes dissolution of iron mono-sulfides (FeS) and calcium carbonates ($CaCO_3$). The hydrogen sulfide ($H_2S$) produced by iron mono-sulfide dissolution is used by the cable bacteria. The $Fe^{2+}$ diffuses both upward and downward. When $Fe^{2+}$ encounters hydrogen sulfide in the sulfidic zone it precipitates again as iron(II)sulfide. In the oxic zone, $Fe^{2+}$ is re-oxidized to $Fe^{3+}$, which will then precipitate as amorphous iron oxide. The calcium-ions released by carbonate dissolution diffuse upward and precipitate again as carbonates in the oxic zone where the pH is higher.





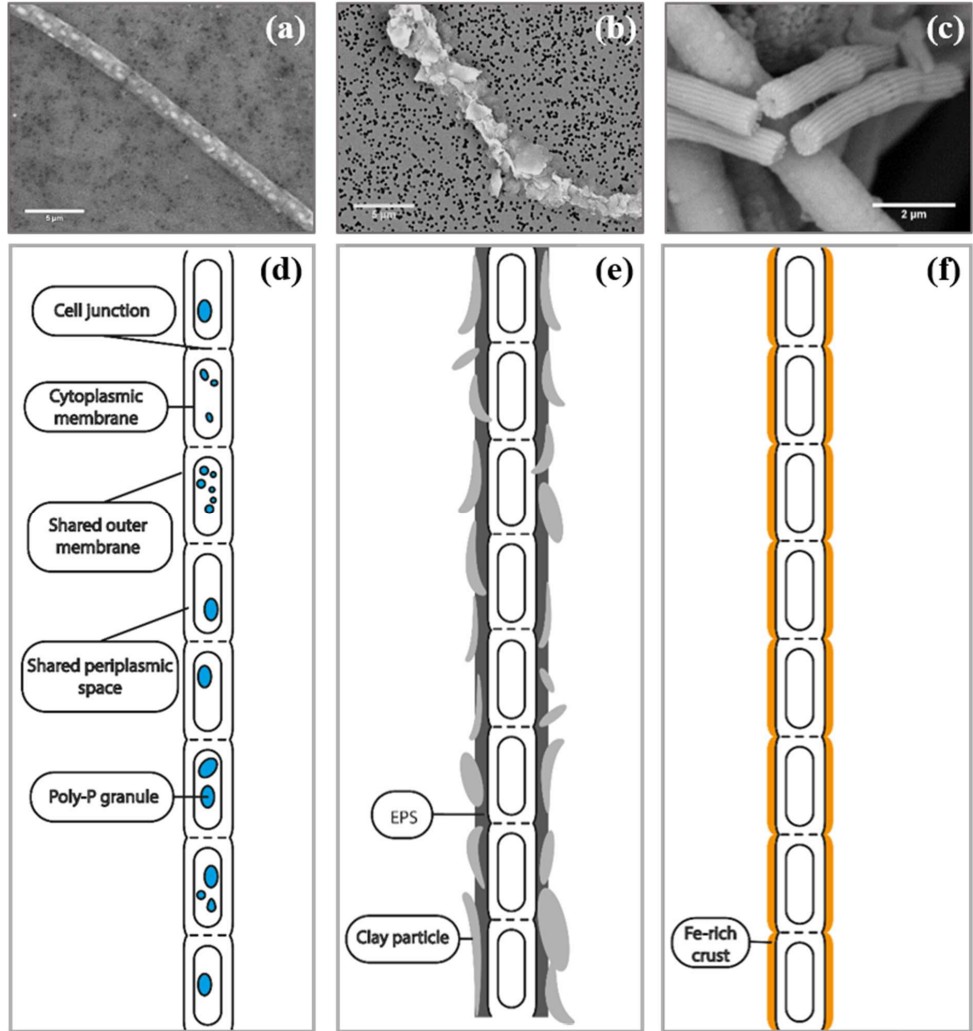

**Figure 2:** SEM images of air-dried cable bacteria and the corresponding conceptual representation exemplifying the different types of mineralization observed in cable bacteria: (**a**) Cable bacterium filament containing polyphosphate granules, (**b**) cable bacterium filament with clay particles attached to it, (**c**) completely encrusted cable bacterium filament that has been broken off at a cell junction. Scale bars represent 5 µm (**a, b**) and 2 µm (**c**). (**d**) The poly-P granules are found within the cells. (**e**) Clay particles are attached to the cable bacteria, most likely due to the presence of extracellular polymeric substances (EPS). (**f**) Cell encrustation happens along the length of the cable bacterium filament thereby preserving the fiber structure.



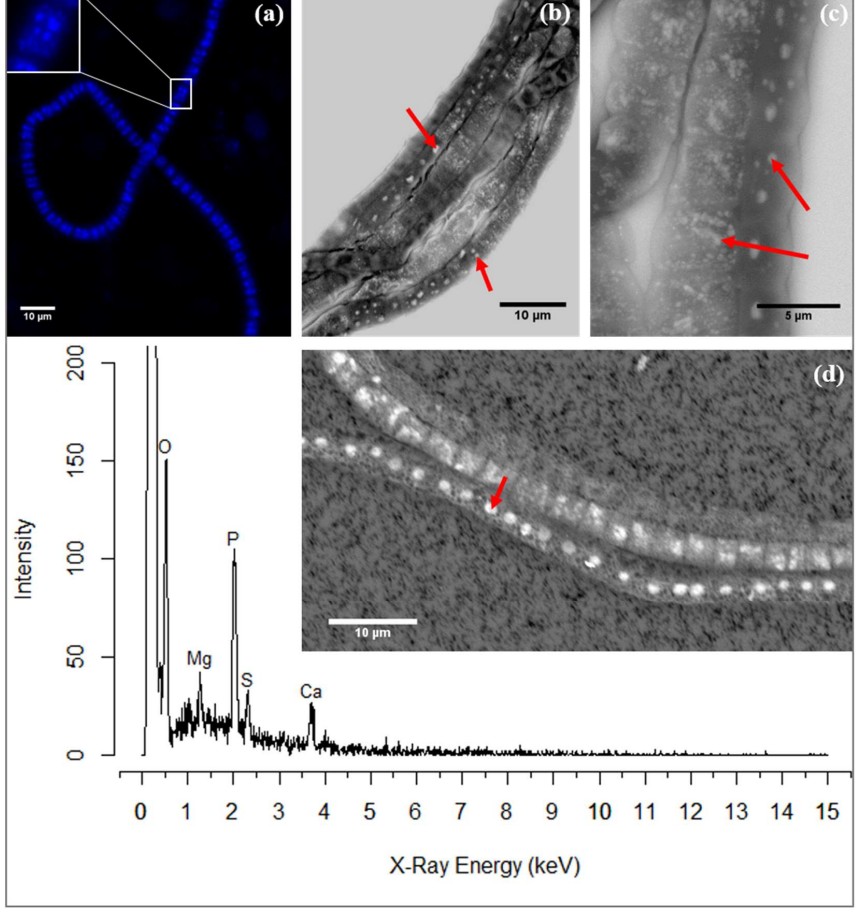

**Figure 3:** (**a**) Fluorescence image of a cable bacterium filament stained with DAPI. The bright blue dots in the cytoplasm are poly-P granules. The inset shows a close-up of a single cell where the poly-P granules are distinguishably visible. (**b**) Scanning electron microscope (SEM) image of a bundle of cable bacteria filaments showing variability in patterns of poly-P granules in filaments extracted from the same environment. The poly-P granules are the bright, white spots indicated by the arrows. (**c**) SEM image of three adjoint cable bacteria filaments showing different sizes and patterns of poly-P granules. Inclusions in image B and C are marked with a red arrow. (**d**) A representative example of an EDS spot analysis of a poly-P granule, where the resultant spectrum shows a high abundance of the elements O, Mg, P, Ca and S. The point of the red arrow on the SEM image indicates the location of the spot analysis. Scale bars represent 10 µm (**a, b, d**) and 5 µm (**c**).



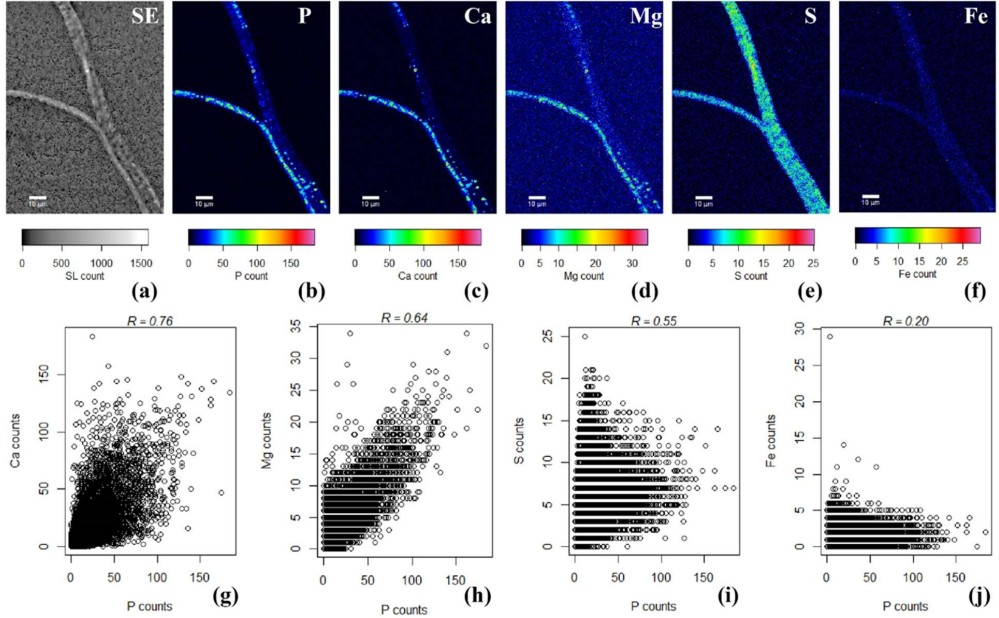

**Figure 4:** False-colored WDS element maps of (**a**) secondary electrons, (**b**) phosphorus, (**c**) calcium, (**d**) magnesium, (**e**) sulfur and (**f**) iron. Scale bars represent 10 μm. To assess the correlation between the elements, scatter plots of (**g**) P and Ca, (**h**) P and Mg, (**i**) and (**j**) P and Fe are shown. All scatterplots were produced from a pixel-by-pixel analysis from the WDS X-ray element maps collected with the EPMA. The Pearson's correlation coefficient (R) values are depicted on top of the scatterplots.



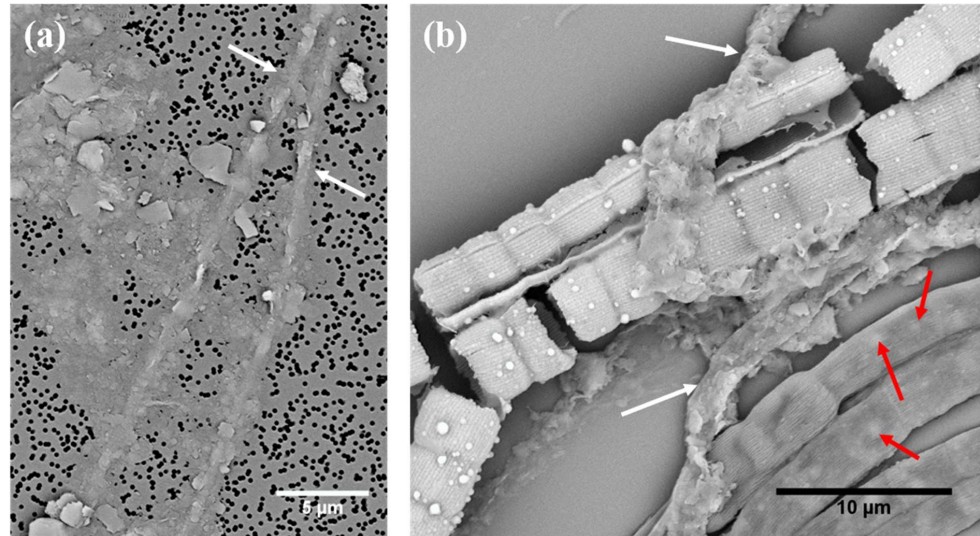

**Figure 5:** (**a**) SEM micrograph of two cable bacteria (white arrows) that have clay particles attached to their surface. (**b**) SEM micrograph showing that three types of mineralization patterns occur within the same environment: cable bacteria that are not encrusted and contain poly-P inclusions (lower right of image, several inclusions are marked with a red arrow), cable bacteria that show extracellular attachment of clay particles (white arrows), and two encrusted cable bacteria that have broken up during the SEM imaging process. Scale bars represent 5 µm (**a**) and 10 µm (**b**).




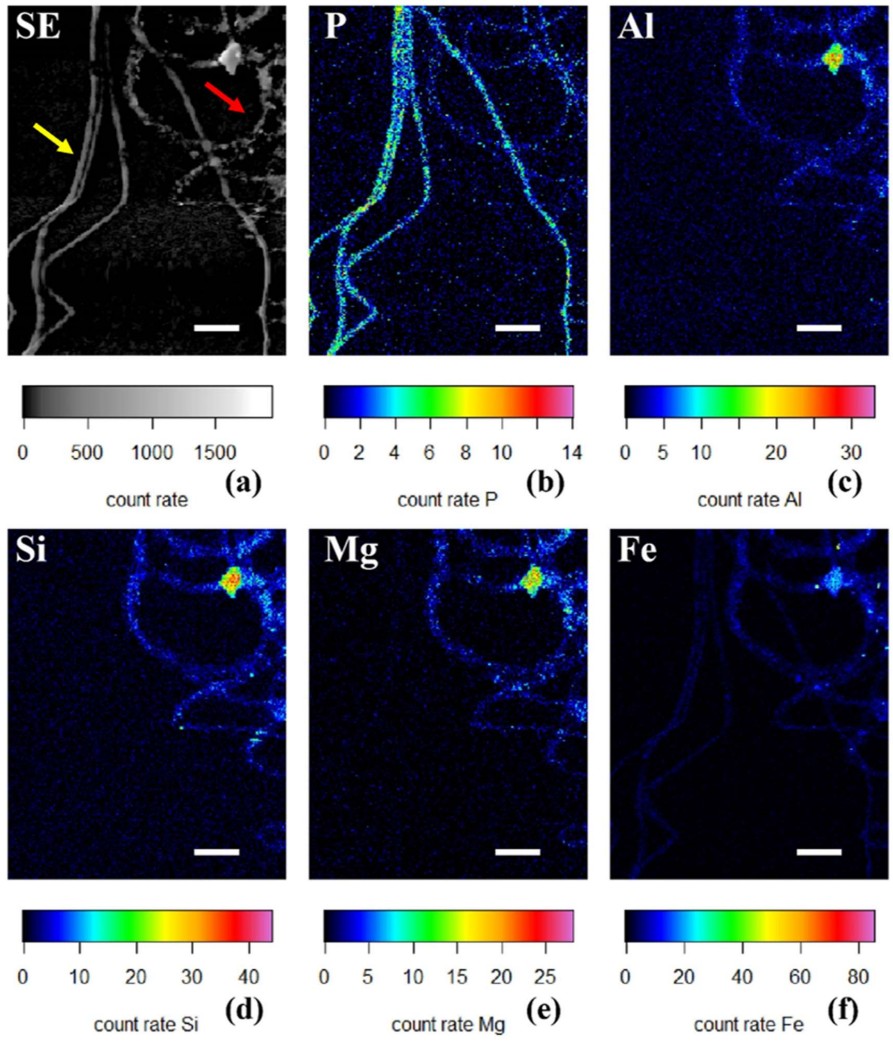

**Figure 6:** (**a**) SEM image in which the yellow arrow points towards cable bacteria without any external attachment of particles while the red arrow points towards cable bacteria that have extracellular attachment of clay particles. False colored WDS element maps of (**b**) phosphorus, (**e**) magnesium and (**f**) iron, and EDS element maps of (**c**) aluminum and (**d**) silicon. Scale bars represent 5 μm.



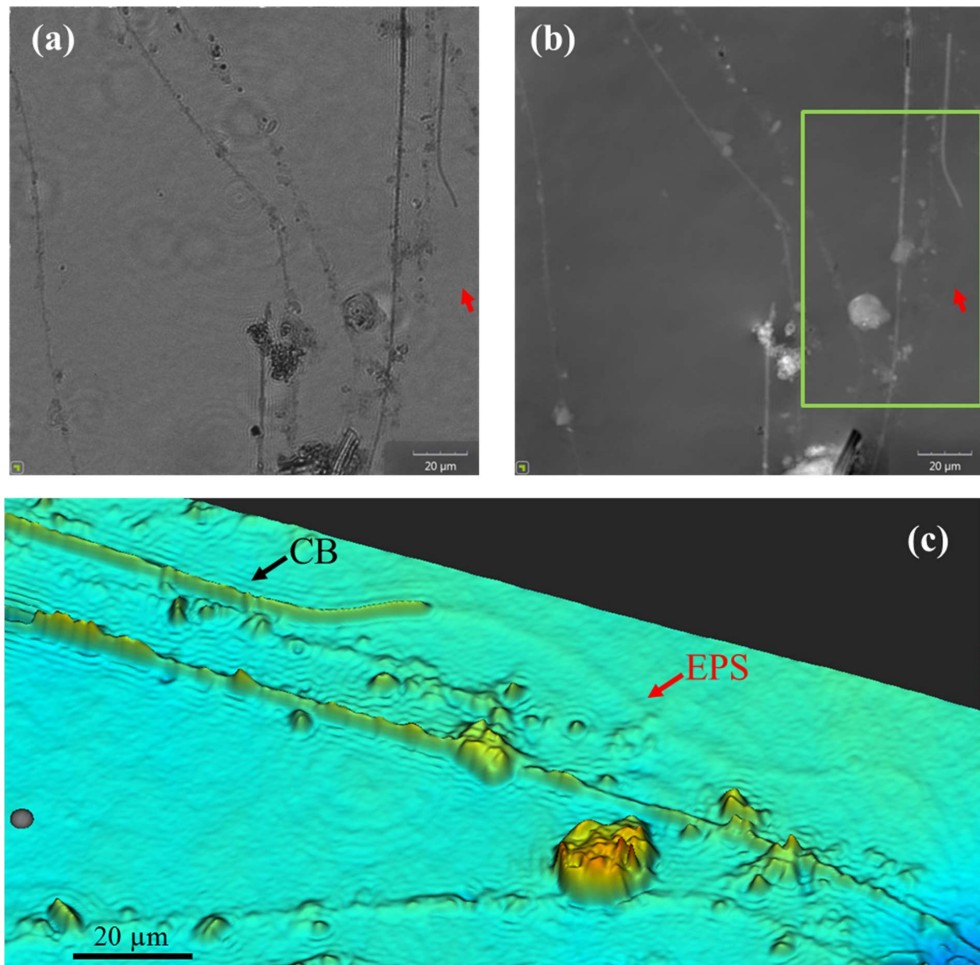


**Figure 7:** Images of cable bacteria grown on glass slides in sediment from the Dutch Marine Lake Grevelingen were captured with an digital holographic microscope. (**a**) The light intensity image, (**b**) the phase image and (**c**) a false color view of a detailed subarea (green box) of the phase image **b**. In the images, the presence of living cable bacteria (CB), as well as the remnants of filaments which had been coated with particles and where most likely only a sheath of extracellular polymeric

substances (EPS) remains, can be observed.




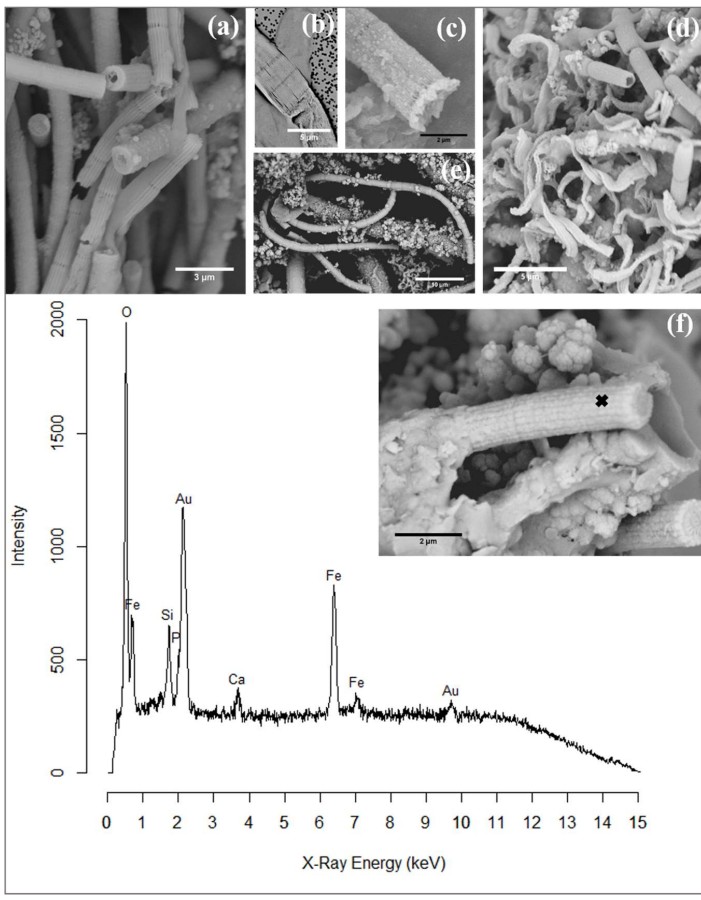

**Figure 8:** SEM images of cell encrustation in cable bacteria: (**a**) Sediment from an incubated core from Mokbaai after 38 days showing different degrees of encrustation on the cable bacteria. (**b**) Cells of a cable bacteria filament showing encrustation of the filament ridges. (**c**) Mineralized stalk that has been broken at a cell-cell junction showing that iron(oxyhydr)oxide has precipitated at the cell-cell junction. It also shows the presence of nanometer sized globules. (**d**) sediment from the oxic layer from Mokbaai after 24 days showing encrusted cable bacteria as well as the twisted stalks from the Fe(II) oxidizing *Gallionella* spp. (**e**) Sediment from an incubated core from the black Sea showing a large part of a looping encrusted cable bacteria filament with attached minerals. (**f**) Broken mineralized remains of a cable bacterium showing the presence of the ridges of the cell and the presence of nanometer sized globules. (**g**) A representative example of a composition spectrum from an EDS spot analysis of encrustation observed on a cable bacteria. The spectrum shows the presence of O, Fe, Si, and Ca in the encrustation. The sample was gold-coated prior to analysis, hence the gold (Au) peak. The spot location is indicated by the black cross in the inset. Scale bars represent 10 µm (**e**), 5 µm (**b, d**), 3 µm (**a**) and 2 µm (**c, f**).




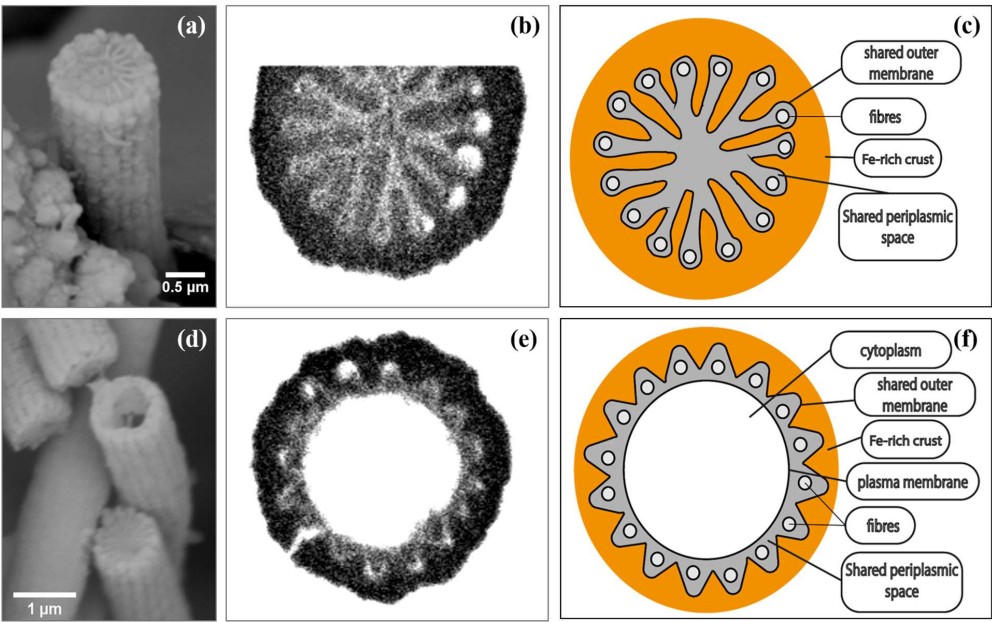

**Figure 9:** SEM images (**a, d**), FIB-SEM images (**b, e**) and conceptual representations (**c, f**) of horizontal cross-sections from cell junctions (**a-c**) and cells (**d-f**) of cable bacteria filaments. (**a**) A mineralized cell junction from a cable bacterium. (**b**) FIB-SEM image of a cell junction showing the connection of ridges at the junctions between cells. (**c**) Conceptual representation of a mineralized cell junction showing that the encrustation is on the outside of the shared outer membrane. (**d**) Two broken cable bacteria filaments where the upper filament shows the inside of a cell while the lower filament shows the (mineralized)
inside at a cell-cell junction. (**e**) FIB-SEM image of a cross-section of a cell and (**f**) the conceptual representation showing that the encrustation is on the outside of the plasma membrane. Scale bars represent 0.5 µm (**a**) and 1 µm (**d**).





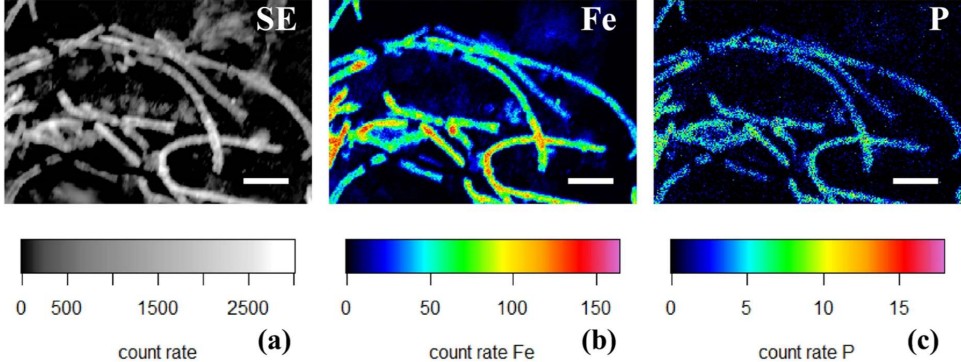

**Figure 10:** (**a**) SEM image and false-colored WDS element maps of the same area for (**b**) phosphorus and (**c**) iron for a consortium of encrusted cable bacteria. Scale bars represent 5 μm.





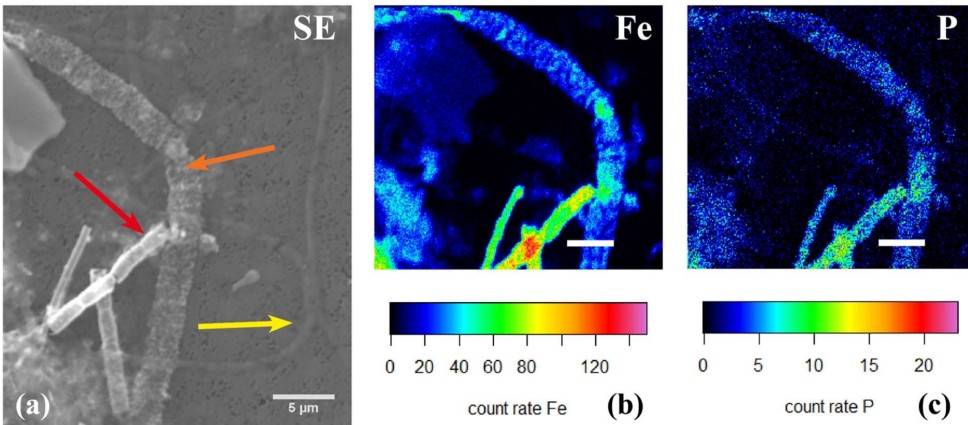

**Figure 11:** (**a**) SEM image with several cable bacteria showing varying degrees of encrustation and the correlated false colored WDS images of the same area showing the presence of phosphorus (**b**) and iron (**c**). The yellow arrow indicates a thin cable bacteria showing no signs of encrustation, the orange arrow a large cable bacteria covered in nanometer-sized globules that contained Fe and P, and the red arrow an encrusted cable bacteria where the crust was mainly composed of Fe and P. Scale bars represent 10 μm (**a**) and 5 μm (**b, c**)





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
