# Peer review of "Mineral formation induced by cable bacteria performing longdistance electron transport in marine sediments"

_Biogeosciences, 2018_

## Referee Comment (RC1) · Anonymous Referee #1 · 15 Nov 2018

Review of manuscript bg-2018-444
Mineral formation induced by cable bacteria performing long-distance electron transport in marine sediments.

General comment:
This manuscript entitled "Mineral formation induced by cable bacteria performing long-distance electron transport in marine sediments" investigates the nature of minerals associated with cable bacteria from different marine settings. Microscopy and spectroscopy analyses reveal (1) intracellular poly-phosphate accumulations in "naked" bacteria, (2) association of cells with extracellular clay minerals, most probably mediated by EPS and (3) different levels of cellular encrustation by Fe-oxyhydroxides adsorbing phosphate.
The methodology is of quality, and the results are well discussed, without any excessive claim. These results are of high importance, as no previous study had been devoted to the analysis of minerals associated with these increasingly studied cable bacteria. They open wide and important issues to be explored. The manuscript is very well written, clearly organized and figures are of high quality. I recommend the publication of this manuscript in Biogeoscience, and have only minor comments.

Specific comments:

- In the introduction, you mention the fact that BIM leads to the formation of minerals that are indistinguishable of abiotic counterparts (l. 88-90). I would temper this claim, as actually, the simple fact that bacteria provide nucleation templates for mineral precipitation can lead to specific mineral textures (e.g. Mirvaux et al 2016) or even initiate mineral precipitation under conditions where this precipitation is kinetically hindered.
- I suggest to remove words like "unintended" (l. 86) or "inadvertent" (l. 538) that are not appropriate for the study of bacteria.
- I suggest not to refer to poly-phosphate as a "mineral".
- I am not sure (unless for the study of iron encrustation) which conditions the samples were exposed to (oxic vs. anoxic). This deserves to be indicated as you prepared samples for microscopy through rinsing in non-degassed (i.e. oxic) water. In case some samples were intially under anoxic conditions, this method would likely induce the formation of secondary minerals (e.g. Fe oxy-hydroxides). Please elucidate this point. In complement to this remark, I suggest the following modifications/discussions to be included:
    o Add the site of collection and depth/condition (oxic vs. anoxic) in each figure legend.
    o I wonder whether you observed different proportions of polyphosphates in samples collected in oxic vs. anoxic zones? This would be interesting to discuss if you have these informations, as part of a potential contribution of cable bacteria to the phosphorus cycle.
- Discussion:
    o Around l. 420: regarding the possibility to evaluate the similarity between poly-P granules and acidocalcisomes, you could also mention the use of fluorescent pH-probes as a potential method.
    o l. 467: regarding acidic micro-environments, you could instead or in addition cite:(Hegler *et al.*, 2010).

- L. 474 and l. 543: For the discussion about the impact of encrustation on metabolic activity, (Miot *et al.*, 2015) have quantified the impact of the level of Fe(III)-mineral encrustation on the uptake or organic molecules (acetate) in Fe(II)-oxidizing bacteria (BoFeN1). This should be discussed here. As an additional strategy to avoid cell encrustation (at the population level), you can thus mention as well the co-existence of bacteria at different levels of encrustation with the naked bacteria. Indeed, you mention that you systematically observe non-encrusted cells, which is consistent with observations by Miot et al (2015) with Fe(II)-oxidizing bacteria.
- L. 502-503: the nature of the minerals could be indeed evaluated by STXM, but also by TEM.
- The FIB-SEM images/videos are very impressive! However, you should mention somewhere in the discussion the potential artifacts that may be induced by the preparation (resin embedding) and analysis of the sample (e.g.(Dohnalkova *et al.*, 2011; Miot *et al.*, 2011; Bassim *et al.*, 2012). Cryo-methods could very interstingly complement your observations.
- You could summarize more clearly the potential role of cable bacteria in Fe-mineral formation: (1) their surface can provide a nucleation site for mineral precipitation, (2) their metabolism (in particular $O_2$ respiration in the oxic zone) may locally increase the pH (local pH gradient around the cells), which would be favorable to Fe-mineral precipitation and growth. The nano-sized globules observed on some cells could correspond to early stages of cell encrustation.

Minor corrections:
- Replace the wording "microscopic techniques" by "microscopy techniques" (e.g. l. 21, 103)
- Material and methods (l. 215 and next): indicate the modes of SEM imaging (backscattered vs secondary electron mode). This should be mentioned in figure legends as well if different modes have been applied.
- Fig. 4: mistake in the legend. Add (i): P and S ...
- L. 400-401: "The storage of P [...] the compounds". I do not understand this sentence.

Bassim ND, De Gregorio BT, Kilcoyne ALD, Scott K, Chou T, Wirick S, Cody G, Stroud RM (2012) Minimizing damage during FIB sample preparation of soft materials: FIB SAMPLE PREPARATION OF SOFT MATERIALS. *Journal of Microscopy* **245**, 288–301.

Dohnalkova AC, Marshall MJ, Arey BW, Williams KH, Buck EC, Fredrickson JK (2011) Imaging Hydrated Microbial Extracellular Polymers: Comparative Analysis by Electron Microscopy. *Applied and Environmental Microbiology* **77**, 1254–1262.

Hegler F, Schmidt C, Schwarz H, Kappler A (2010) Does a low-pH microenvironment around phototrophic FeII-oxidizing bacteria prevent cell encrustation by FeIII

minerals?: Low-pH microenvironment prevents cell encrustation. *FEMS Microbiology Ecology* **74**, 592–600.

Miot J, Maclellan K, Benzerara K, Boisset N (2011) Preservation of protein globules and peptidoglycan in the mineralized cell wall of nitrate-reducing, iron(II)-oxidizing bacteria: a cryo-electron microscopy study: Persistence of organics in mineralized Fe-oxidizing bacteria. *Geobiology* **9**, 459–470.

Miot J, Remusat L, Duprat E, Gonzalez A, Pont S, Poinsot M (2015) Fe biomineralization mirrors individual metabolic activity in a nitrate-dependent Fe(II)-oxidizer. *Frontiers in Microbiology* **6**.

---

## Referee Comment (RC2) · Anonymous Referee #2 · 13 Dec 2018

The manuscript of addresses bio mineralization associated with cable bacteria, a newly discovered group of filamentous bacteria within the Desulfobulbacea family that performs electrogenic sulfur oxidation. Using a series of advanced microscopic techniques, the authors investigate the minerals formed within cells in the filaments or on the exterior of cabel bacteria harvested from sediment-based enrichment cultures. The authors identify the presence of polyphosphates within the cells, the presence of external coatings composed by EPS and (probably) clay minerals; and the presence of external encrustations of iron oxy hydroxides. The findings are discussed primarily in relation to the eco-physiology of cable bacteria, which makes the paper relevant for the community of researchers dealing with theses aspects. However, the work also

addresses more general aspects of biominralization, through the focus on model organisms, which through its peculiar metabolism has significant influence on many geochemical pathways. More over this organism has been shown to be abundant in many aquatic sediments and therefore I think that the study would be relevant and interesting also for readers of BG, that are not directly associated to cable bacteria research. In general, I find the manuscript well prepared, with (mostly) proper citing (see my point 3) and credit to related work. To my knowledge, the methodology used is sound and I find no reason to doubt the results and the interpretations of raw data. I therefore recommend publication of the manuscript.

I have a few suggestions and questions for the authors to consider. 1. When discussing of encrustation the authors refer to the LPS of gram-negative bacteria, as the cable bacteria are gram-negative bacteria. My question to this end is what is known about the composition of the outer membrane of cable bacteria i.e. the common membrane that encapsulates all individual cells in the filament? Is there any evidence that this membrane is composed from LPS? Note that we can easily imagine that the individual cells in the filament has both an inner and an outer membrane composed as for gram-negative prokaryotes. , and that the common outer membrane is composed differently than from that?; Until more knowledge about the composition of the outer membrane is known, I do not think that the authors cannot make firm conclusions about the relationship between iron precipitations and the membrane properties and I encourage the authors to tone this discussion down. 2. In the discussion on the mechanism behind the formation of iron (oxy) hydrates encrustations on the cable bacteria the work of Otte et al. 2018 is used as a model for explanation of the crust formation. This model assumes direct electron transfer between cable bacteria and iron oxidizers present in anoxic sediment strata and as a consequence formation of iron (oxy) hydrates in the absence of oxygen. Perhaps this can occur but is really documented suffienctly well to be used as an explanation for the observation that some cable bacteria are covered with iron (oxy) hydrates? I do not think so. More over as I read the methods section ,cable bacteria for this analysis were collected from the oxic zone of the sediments

and there you do not need anything more than well known geochemistry to explain the formation iron (oxy) hydrates. So I suggest that the author tone down the more exotic explanations and choose the most simple model: that the iron (oxy) hydrates are formed through well-known reactions between O2 and Fe2+ in the oxic zone. 3. There are some references to unpublished work (e.g. Cornelissen. subm. ) and I suggest that these are taken out of the manuscript. In my view the information the Cornelissen. et al. subm. Paper, as cited in the manuscript does not contribute to an understanding of the data as it apparently deals with the internal structure of cable bacteria. Encrustation (the topic of the paragraph) is related to the external structure – i.e. the outer membrane. Please also be aware that all information related to this is sufficiently well described in the Pfeffer et al 2012 paper, and that the Meysman 2018 paper, which also is cited along the line of description of the cellular structures (l.453) does appear in the reference list. Here only Meysman 2017 appears and this is a review that does not add more information to the topic, than already described in the primary literature.

———————————————————————

---

## Author Comment (AC1) · 21 Dec 2018

**Response to Referee #1**

**General comment:**
This manuscript entitled "Mineral formation induced by cable bacteria performing long-distance electron transport in marine sediments" investigates the nature of minerals associated with cable bacteria from different marine settings. Microscopy and spectroscopy analyses reveal (1) intracellular poly-phosphate accumulations in "naked" bacteria, (2) association of cells with extracellular clay minerals, most probably mediated by EPS and (3) different levels of encrustation by Fe-oxyhydroxides adsorbing phosphate.

The methodology is of quality, and the results are well discussed, without any excessive claim. These results are of high importance, as no previous study has been devoted to the analysis of minerals associated with these increasingly studied cable bacteria. They open wide and important issues to be explored. The manuscript is very well written, clearly organized and figures are of high quality. I recommend the publication of this manuscript in Biogeoscience, and have only minor comments.

*Answer to general comment*

We would like to thank this referee for the positive input and the very detailed comments which made it possible for us to dot the i's and cross the t's. We reply to each specific comment below.

**Comment #1**

In the introduction, you mention the fact that BIM leads to the formation of minerals that are indistinguishable of abiotic counterparts (l. 88-90). I would temper this claim, as actually, the simple fact that bacteria provide nucleation templates for mineral precipitation can lead to specific mineral textures (e.g. Mirvaux et al 2016) or even initiate mineral precipitation under conditions where this precipitation is kinetically hindered.

*Answer to comment #1*

We were not aware of this research, and we fully agree with the remark. We have changed the text as follows:

"Minerals that form by BIM generally nucleate and grow extracellularly as a result of the metabolic activity of the organism and subsequent chemical reactions involving metabolic by-products. BIM is an uncontrolled consequence of metabolic activity. The minerals formed are generally characterized by poor crystallinity, broad particle-size distributions, and lack of specific crystal morphology (Lowenstam and Weiner, 1989; Frankel and Bazylinsky, 2003). Both abiotic precipitation and BIM may result in minerals that are chemically and morphologically similar, though in other cases, there may be morphological differences. . This is because the bacterial surface provides nucleation templates for mineral precipitation, which act as a template for the growth and organization of the precipitated particles, thus leading to specific mineral textures (Mirvaux et al., 2016), or bacteria may initiate mineral precipitation under conditions where abiotic precipitation is kinetically hindered, which may also steer mineral morphology."

**Comment #2**

I suggest to remove words like "unintended" (l. 86) or "inadvertent" (l. 538) that are not appropriate for the study of bacteria.

*Answer to comment #2*

In the revised text "unintended" is removed and "inadvertent" is changed to "uncontrolled".

**Comment #3**

I suggest not to refer to poly-phosphate as a "mineral".

*Answer to comment #3*

We agree with this comment and will change the abstract to "…observed the formation of polyphosphate granules within the cells and two different types of mineral formation directly associated with multi-cellular filaments of these cable bacteria: the attachment of clay-particles into a coating surrounding the bacteria, and encrustation of the cell envelope by iron minerals."

**Comment #4**

I am not sure (unless for the study of iron encrustation) which conditions the samples were exposed to (oxic vs. anoxic). This deserves to be indicated as you prepared samples for microscopy through rinsing in non-degassed (i.e. oxic) water. In case some samples were initially under anoxic conditions, this method would likely induce the formation of secondary minerals (e.g. Fe oxyhydroxides). Please elucidate this point. In complement to this remark, I suggest the following modifications/discussions to be included:
   o   Add the site of collection and depth/condition (oxic vs. anoxic) in each figure legend.
   o   I wonder whether you observed different proportions of polyphosphates in samples collected in oxic vs. anoxic zones? This would be interesting to discuss if you have these informations, as part of a potential contribution of cable bacteria to the phosphorus cycle.

*Answer to comment #4*

This is a valuable remark, and we have will adapt the text to clarify differences (if any) between oxic vs. anoxic zones, and potential impacts of filament preparation.

In the many samples that we screened, there was no observation of Fe oxyhydroxides in samples from the anoxic zone, and so it is safe to assume that secondary Fe mineral formation did not occur despite the oxic conditions during sample preparation. Because individual cable bacteria or clumps of cable bacteria are washed through drops of de-ionized water there are no ions available for (secondary) mineral formation (e.g. there is ferrous iron to be oxidized). Only filaments extracted from the oxic zone showed Fe encrustation. This Fe encrustation is already observed under a light microscope right after the filaments are extracted from the sediment (before being rinsed with non-degassed water), and so the encrustation is formed *in situ*.

The sheaths with extracellular clay minerals observed around filaments were observed in both the oxic and suboxic zone and thus appears to form independent from the redox zonation. This sheath is already observed (albeit not as clearly as with electron microscopy) under a light microscope when the filaments are extracted from the sediment before being rinsed with non-degassed water.
We are aware that mineralogy might change during sample preparation. For now, we did not look at the mineral structure but elemental composition. When investigating the mineral structure (e.g. with STXM) this needs to be taken into account.

Concerning the polyphosphate granules we have only qualitative data, so for now we cannot elucidate if there are different proportions of polyphosphates in samples collected in the oxic vs. anoxic zones. To get more quantitative data nanoSIMS measurements or other small-scale techniques need to be performed that are not a part of this manuscript. The potential contribution of polyphosphates in cable bacteria to the phosphorus cycle has been investigated for Marine Lake Grevelingen and the

contribution was found to be negligible (Sulu-Gambari et al., 2016). Research on a more spatial and temporal scale would be useful but is not a part of this manuscript.

For clarity, we added the site of collection and the condition from where the sample was taken to the legends.

**Comment #5**

Discussion:

- Around l. 420: regarding the possibility to evaluate the similarity between poly-P granules and acidocalcisomes, you could also mention the use of fluorescent pH-probes as a potential method.
- L. 467: regarding acidic micro-environments, you could instead or in addition cite: (Hegler *et al.*, 2010).
- L. 474 and l. 543: For the discussion about the impact of encrustation on metabolic activity, (Miot *et al.*, 2015) have quantified the impact of the level of Fe(III)-mineral encrustation on the uptake of organic molecules (acetate) in Fe(II)-oxidizing bacteria (BoFeN1). This should be discussed here. As an additional strategy to avoid cell encrustation (at the population level), you can thus mention the co-existence of bacteria at different levels of encrustation with the naked cable bacteria. Indeed, you mention that you systematically observe non-encrusted cells, which is consistent with observations by Miot et al (2015) with Fe(II)-oxidizing bacteria.
- L. 502-503: the nature of the minerals could be indeed evaluated by STXM, but also by TEM.
- The FIB-SEM images/videos are very impressive! However, you should mention somewhere in the discussion the potential artifacts that may be induced by the preparation (resin embedding) and analysis of the sample (e.g. (Dohnalkova *et al.*, 2011; Miot *et al.,* 2011; Bassim *et al.*, 2012). Cryo-methods could very interestingly complement your observations.
- You could summarize more clearly the potential role of cable bacteria in Fe-mineral formation: (1) their surface can provide a nucleation site for mineral precipitation, (2) their metabolism (in particular $O_2$ respiration in the oxic zone) may locally increase the pH (local pH gradient around the cells), which would be favorable to Fe-mineral precipitation and growth. The nano-sized globules observed on some cells could correspond to early stages of cell encrustation.

*Answer to comment #5*

- We will add the use of fluorescent pH-probe as a potential method and use Brock et al., 2012 and Hegler et al, 2010 as a reference.
- Hegler et al, 2010 has now been cited in addition to Schadler et al, 2009. Thank you. We were not aware of this publication.
- We will add a sentence after l. 474:
- "Cell encrustation could potentially limit the diffusion of substrates and nutrients to the cell, impair uptake of these compounds across the membrane, and as a consequence lead to the stagnation of cell metabolism and eventually even to cell death (Konhauser, 1998b; Schädler et al., 2009). When a culture of the Fe(II) oxidizing bacteria *Acidovorax* sp. strain Bo1FeN1 was exposed to high concentrations of $Fe^{2+}$, most cells became encrusted with iron minerals. Cells that were moderately encrusted still had the capacity to assimilate acetate but with increasing levels of iron encrustation the capacity to assimilate carbon decreased exponentially. Remarkably, a small proportion of cells remained free of encrustation and metabolically active implying that phenotypic heterogeneity might be a viable strategy to cope with biomineralization (Miot et al., 2015). Since both encrusted and non-encrusted filaments co-exist (Fig. 11a) this strategy might also be employed by cable bacteria."

- TEM was added as a method.
- We are aware that the sample preparation method may have resulted in the loss of microstructures and a change in cell morphology. However, the FIB-SEM videos were used to analyze the thickness and structure of the encrustation and to identify the location of the biomineral layer, which could be on the outside of the filament , or alternatively, within the periplasmic space underneath the ridge structure. In the FIB-SEM images/videos periplasmic space is visible and therefore the possible loss of microstructures or changes in the chemical composition would not change the conclusion that the biomineral formation took place at the outside of the mineral structure. Potential artifacts would be a problem when looking at the chemical interaction between the outer cell surface, EPS and the mineral layer. For this, cryogenic methods appear to be the most promising since they would preserve the native structure. As a suggestion for further research we have added a sentence after l. 506 where cryogenic methods are mentioned and the suggested publications were cited:
  "To further investigate the interaction between the outer cell surface, EPS and the mineral layer, cryogenic methods appear to be promising since they would preserve the native structure (Bassim et al., 2012; Dohnalkove et al., 2011; Miot et al., 2011b)"
- To articulate the potential role of cable bacteria in Fe biomineral formation we will rewrite the last paragraph of the discussion as follows:
  "It appears that the metabolism of cable bacteria results in a cascade of reactions that eventually results in the uncontrolled mineralization of filaments that are present in the oxic zone. The cell surface provides a nucleation site and template for mineral formation, and the increase of the pH in the oxic zone as a result of the electrogenic metabolism of cable bacteria, favors Fe-mineral precipitation and growth. Since the mineral precipation does not appear to be controlled by the cable bacteria, it forms an example of biologically induced mineralization. The formation of a mineral crust on a cell surface could potentially limit cell metabolism and may eventually lead to cell death (Konhauser, 1998a; Schädler et al., 2009). However, the extent to which this affects cable bacteria is currently unknown and so the impact of encrustation on cable bacteria metabolism needs to be resolved."

**Minor corrections:**

- Replace the wording "microscopic techniques" by "microscopy techniques" (e.g. l. 21, 103)
- Materials and methods (l. 215 and next): indicate the modes of SEM imaging (backscattered vs secondary electron mode). This should be mentioned in figure legends as well if different modes have been applied.
- Fig. 4: mistake in the legend. Add (i): P and S…..
- L. 400-401: "The storage of P […] the compounds". I do not understand this sentence.

*Answer to minor corrections:*

All minor corrections will be changed in the final manuscript. We would like to thank the reviewer for the attention to detail.

L. 400-401 will be rewritten and incorporated with l. 393-395: "Acidocalcisomes are an electron-dense, acidic compartment containing a matrix of pyrophosphate and polyphosphates with bound calcium and other cations, mainly magnesium and potassium. The formation of acidocalcisomes hence allows increased uptake of both phosphorus compounds and cations (Docampo and Moreno, 2012)."

---

## Author Comment (AC2) · 21 Dec 2018

**Response to Referee #2**

**General comments**

The manuscript addresses bio mineralization associated with cable bacteria, a newly discovered group of filamentous bacteria within the *Desulfobulbaceae* family that performs electrogenic sulfide oxidation. Using a series of advanced microscopic techniques, the authors investigate the minerals formed within cells in the filaments or on the exterior of cable bacteria harvested from sediment-based enrichment culture. The authors identify the presence of polyphosphates within the cells, the presence of external coatings composed of EPS and (probably) clay minerals; and the presence of external encrustations of iron oxy hydroxides. The findings are discussed primarily in relation to the eco-physiology of cable bacteria, which makes the paper relevant for the community of researchers dealing with these aspects. However, the work also addresses more general aspects of biomineralization, through the focus on model organisms, which through its peculiar metabolism has significant influence on many geochemical pathways. Moreover, this organism has been shown to be abundant in many aquatic sediments and therefore I think that the study would be relevant and interesting also for readers of BG that are not directly associated to cable bacteria research. In general, I find the manuscript well prepared, with (mostly) proper citing (see my point 3) and credit to related work. To my knowledge, the methodology used is sound and I find no reason to doubt the results and the interpretations of raw data. I therefore recommend publication of this manuscript.

*Answer to general comments*

We would like to thank the author for his/her attention to detail and the positive recommendation Each of his/her suggestions is discussed individually below.

**Comment 1**

When discussing the encrustation the authors refer to the LPS of gram-negative bacteria, as the cable bacteria are gram-negative bacteria. My question to this end is what is known about the composition of the outer membrane of cable bacteria i.e. the common membrane that encapsulates all individual cells in the filament? Is there any evidence that this membrane is composed from LPS? Note that we can easily image that the individual cells in the filament has both an inner and an outer membrane composed as for gram-negative prokaryotes and that the common outer membrane is composed differently than from that? Until more knowledge about the composition of the outer membrane is known, I do not think that the authors cannot make firm conclusions about the relationship between iron precipitations and the membrane properties and I encourage the authors to tone it down.

*Answer to comment 1*

At the moment not much is known about the chemical structure of the common membrane of cable bacteria and we do not have any evidence that this membrane is composed of LPS. So this needs to be investigated. However, in our manuscript, no claims or conclusions are being made about the relationship between iron precipitation and membrane properties (such as the presence of LPS). We do, however, present a hypothesis that might explain the precipitation of iron minerals containing Ca, Mg and Si that was observed and we believe it is a valid hypothesis in this context.

To clarify to the readers that no research has been done on the chemical composition of the outer membrane, a sentence will be added to the paragraph that starts at L.476:

"Although the structure and composition of the common outer membrane shared by all cells in a cable bacterium filament is not known, gram-negative bacteria typically possess an outer membrane that

forms an asymmetric lipid-protein bilayer which incorporates phospholipids, lipopolysaccharides (LPS) and proteins, and separates the external environment from the periplasm."

We will temper our claim by not mentioning phosphoryl and carboxyl surface groups in the final paragraph of our discussion. It will be rewritten as follows:

"It appears that the metabolism of cable bacteria results in a cascade of reactions that eventually results in the uncontrolled mineralization of filaments that are present in the oxic zone. The cell surface provides a nucleation site and template for mineral formation, and the increase of the pH in the oxic zone as a result of the electrogenic metabolism of cable bacteria, favors Fe-mineral precipitation and growth. Since the mineral precipation does not appear to be controlled by the cable bacteria, it forms an example of biologically induced mineralization. The formation of a mineral crust on a cell surface could potentially limit cell metabolism and may eventually lead to cell death (Konhauser, 1998a; Schädler et al., 2009). However, the extent to which this affects cable bacteria is currently unknown and so the impact of encrustation on cable bacteria metabolism needs to be resolved."

**Comment 2**

In the discussion on the mechanics behind the formation of iron (oxy) hydrates encrustations on the cable bacteria the work of Otte et al. 2018 is used as a model for explanation of crust formation. This model assumes direct electron transfer between cable bacteria and iron oxidizers present in anoxic sediment strata and as a consequence formation of iron (oxy) hydrates in the absence of oxygen. Perhaps this can occur but is really documented sufficiently well to be used as an explanation for the observation that some cable bacteria are covered with iron (oxy) hydrates? I do not think so. More over as I read the methods section, cable bacteria for this analysis were collected from the oxic zone of the sediments and there you do not need anything more than well-known geochemistry to explain the formation of iron (oxy) hydrates. So I suggest that the authors tone down the more exotic explanations and choose the most simple model: that the iron (oxy) hydrates are formed through well-known reactions between O2 and Fe2+ in the oxic zone.

*Answer to comment 2*

We agree fully with the referee on this point. We do not think there is enough evidence for interspecies electron transfer from cable bacteria to iron oxidizers. Effectively, our reference to Otte et al (2018) was intended to signal the co-existence of cable bacteria and iron-oxidizing bacteria, but not intended to support their model of interspecies electron transfer.

We mention first (l. 509 – l. 513):

"Iron (oxyhydr)oxides are widespread and form in any environment where $Fe^{2+}$-bearing waters come into contact with $O_2$ (Konhauser and Riding, 2012). In electrogenic sediments, the formation of an iron oxide crust has been observed in both laboratory experiments (Risgaard-Petersen et al., 2012; Rao et al., 2016) as well as in-situ (Seitaj et al., 2015; Sulu-Gambari et al., 2016). It is an example of biologically induced mineralization (Lowenstam and Weiner, 1989) resulting from the metabolic activity of the cable bacteria and the subsequent availability and re-oxidation of the $Fe^{2+}$ ions in the oxic zone."

Next we discuss whether the oxidation of $Fe^{2+}$ in the **oxic zone** is biotic or abiotic for which we use the co-existence of cable bacteria and iron-oxidizing bacteria found by Otte et al. (2018) as well as the presence of stalks of *Gallionella spp*. as an argument to hypothesize that the oxidation of iron in the oxic zone is most likely biotic. We do not mention any of the models discussed by Otte et al. (2018).

To clarify this, we will rewrite this paragraph as follows:

"At pH values above 8, the oxidation rate is fast, but no longer varies with the pH. The rate of oxidation is both thermodynamically and kinetically enhanced by adsorption of dissolved iron species to hydrous oxide surfaces (Morgan and Lahav, 2007). For the $Fe^{2+}$ oxidation to be biotic, $Fe^{2+}$ oxidizing bacteria need to outcompete the abiotic reaction. The twisted stalks of the $Fe^{2+}$ oxidizing *Gallionella* spp. have been found in samples of encrusted cable bacteria (Fig. 8d) showing that, despite the high pH values in the oxic zone, $Fe^{2+}$ oxidizing bacteria (partly) outcompete the abiotic reaction. There is also direct evidence for the co-existence of active cable bacteria and $Fe^{2+}$ oxidizing and $Fe^{3+}$ reducing bacteria in sediments representative of typical marine environments (Otte et al., 2018). Whenever cable bacteria were abundantly present (0.1%-4.5%), both $Fe^{2+}$ oxidizing and $Fe^{3+}$ reducing bacteria were homogeneously distributed throughout the sediment and their presence was therefore decoupled from the traditional geochemical gradients (Otte et al., 2018). After oxidation, ferric iron (hydr)oxides are expected to precipitate more or less instantly at the alkaline pH values in the oxic zone due to the low solubility of $Fe^{3+}$ under these conditions. The alkaline pH value in the oxic zone is the result of the separation of two redox half-reactions in electrogenic sulfur oxidation by cable bacteria (Fig. 1). Precipitation most likely occurs directly where the $Fe^{3+}$ is formed and because of the proximity of cells, $Fe^{3+}$ ions, $Fe^{3+}$ complexes, $Fe^{3+}$ colloids and $Fe^{3+}$ minerals are expected to adsorb to prokaryotic cell surfaces that are generally effective sorption interfaces for metal ions (Beveridge, 1999; Ferris et al., 1987; Fortin et al., 1997) as well as negatively charged silicate ions (Schultze-Lam et al., 1996)."

**Comment 3**

There are some references to unpublished work (e.g. Cornelissen. subm.) and I suggest that these are taken out of the manuscript. In my view the information the Cornelissen. et al. subm. Paper, as cited in the manuscript does not contribute to an understanding of the data as it apparently deals with the internal structure of the cable bacteria. Encrustation (the topic of the paragraph) is related to external structure – i.e. the outer membrane. Please also be aware that all information related to this is sufficiently well described in the Pfeffer et al 2012 paper, and that the Meysman 2018 paper, which also is cited along the line of description of the cellular structures (l.453) does appear in the reference list. Here only Meysman 2017 appears and this is a review that does not add more information to the topic, than already described in the primary literature.

*Answer to comment 3*

The paper submitted by Cornelissen et al. has now been published in Frontiers in Microbiology, so the reference will be changed in the finalized version of this manuscript. It deals with the structure and morphology of cell envelope, both internal and external, It expands the knowledge that was presented by Pfeffer et al in 2012, by providing a quantitative model of outer surface morphology by means of extensive atomic force microscopy measurements (for different types of cable bacteria). Therefore we believe that a reference to "Cornelissen et al." is justified. We will change all references to "Cornelissen et al." to "Pfeffer et al. and Cornelissen et al.".

The reference to Meysman 2018 will be removed as well as the incorrect reference to Meysman 2017.